# Dust mass, CCN, and INP profiling with polarization lidar: Updated POLIPHON conversion factors from global AERONET analysis

Albert Ansmann[1], Rodanthi-Elisavet Mamouri[2], Julian Hofer[1], Holger Baars[1], Dietrich Althausen[1], and Sabur F. Abdullaev[3]

[1]Leibniz Institute for Tropospheric Research, Leipzig, Germany
[2]Cyprus University of Technology, Dep. of Civil Engineering and Geomatics, Limassol, Cyprus
[3]Physical Technical Institute, Academy of Sciences of the Republic of Tajikistan, Dushanbe, Tajikistan

*Correspondence to:* A. Ansmann
(albert@tropos.de)

**Abstract.** The POLIPHON (Polarization Lidar Photometer Networking) method permits the retrieval of particle number, surface area, and volume concentration for dust and non-dust aerosol components. The obtained microphysical properties are used to estimate height profiles of particle mass, cloud condensation nucleus (CCN) and ice-nucleation particle (INP) concentrations. **The conversion of aerosol-type-dependent particle extinction coefficients, derived from polarization lidar observations,**

**into the aerosol microphysical properties (number, surface area, volume) forms the central part of the POLIPHON computations**. These conversion parameters are determined from Aerosol Robotic Network (AERONET) aerosol climatologies of optical and microphysical properties. In this article, we focus on the dust-related POLIPHON retrieval products and present an extended set of dust conversion factors considering all relevant deserts around the globe. We apply the new conversion factor set to a dust measurement with polarization lidar in Dushanbe, Tajikistan, in central Asia. Strong aerosol layering

was observed with mineral dust advected from Kazakhstan (0-2 km height), Iran (2-5 km), the Arabian peninsula (5-7 km), and the Sahara (8-10 km). POLIPHON results obtained with different sets of conversion parameters were contrasted in this Central Asian case study and permitted an estimation of the conversion uncertainties.

## 1 Introduction

Increasing urbanization, rising aerosol pollution levels, and the need for an improved understanding of the relationship between

aerosols, clouds, and precipitation motivated us to develop a robust and easy-to-handle lidar method for an height-resolved retrieval of particle mass concentration and cloud-relevant parameters (Mamouri and Ansmann, 2016, 2017). Lidar is the only available technique for continuous monitoring and detailed vertical profiling of local and regional aerosol conditions (see, e.g., Baars et al., 2016). The recently introduced POLIPHON (Polarization Lidar Photometer Networking) technique allows the requested aerosol monitoring of environmental and meteorological relevant aerosol properties such as cloud condensation

nucleus (CCN) and ice-nucleating particle (INP) concentrations. **The method combines the unique features of polarization lidar (see, e.g., Freudenthaler et al., 2009; Tesche et al., 2011) and of the well-established global aerosol climatology of aerosol optical and microphysical properties provided by AERONET (Aerosol Robotic Network) (Holben et al., 1998).**

**The polarization lidar technique permits the separation of mineral dust and non-dust aerosol components (such as anthropogenic haze and biomass burning haze over continents), whereas the multi-year AERONET data base allows us to develop climatologically robust relationships between observable aerosol-type-dependent particle optical properties and the desired environmental and cloud-relevant aerosol parameters separately for the basic aerosol types of mineral**

**dust, continental fine-mode aerosol pollution, and marine particles. These two aspects (aerosol type separation and aerosol-type-dependent conversion into respective microphysical properties) are the essential parts of the POLIPHON method which is described in Sect. 2 with focus on mineral dust applications.**

The POLIPHON method can be applied to observations with wide-spread ground-based single-wavelength polarization lidars (Cordoba-Jabonero et al., 2018) as well as to spaceborne single-wavelength polarization lidar measurements with CALIOP

(Cloud-Aerosol Lidar Observations with Orthogonal Polarization) (Winker et al., 2009; Mamouri and Ansmann, 2015; Marinou et al., 2017, 2018). **It is generally applicable also to observations with multiwavelength polarization lidars operated in well-organized ground-based lidar networks such as the European Aerosol Research Lidar Network (EARLINET) (Pappalardo et al., 2014) or the Asian Dust Network (Shimizu et al., 2004).** POLIPHON products have been successfully compared with in situ measured vertical profiles of particle mass concentration (Mamali al., 2018), CCN (Düsing et al., 2018),

and INP concentration (Schrod et al., 2017; Marinou et al., 2018), and recently with fine and coarse dust mass concentration, CCN concentration, and INP-relevant aerosol properties observed with research aircraft (over the lidar site) after long-range transport to the Caribbean (Haarig et al., 2019).

In this article, we extend the POLIPHON method towards global dust applications. This effort is triggered by several reasons: First of all, mineral dust is a global player in the climate system by sensitively influencing the radiative transfer in the Earth

atmosphere and by serving as an important reservoir for favorable INPs (Hoose and Möhler, 2012; Murray et al., 2012; Kanji et al., 2017). Heterogeneous ice nucleation on dust INPs can initiate ice and precipitation formation already at high temperatures of $-15$ to $-35°$C (Seifert et al., 2010). Without aerosol particles ice and rain formation rates would be strongly reduced in the atmosphere. However, and this is the second reason for the dust-related improvements presented here, POLIPHON has, in the majority of case studies, been applied to Saharan dust observations only. Thus, only Saharan-dust-related conversion parameters

have been determined so far. Now the questions arise: Are these Saharan dust conversion factors valid for the different dust regimes around the globe? Do we need different sets of conversion factors for, e.g., Saharan dust, Middle East dust, East Asian dust, dust in North and South America, South Africa, and Australia? In case that the differences in the conversion factors for different dust regions are small we may be able to develop one universal set of conversion factors which would facilitate the use of the POLIPHON method from space significantly? Guided by these questions we studied the AERONET data base

regarding the relationship between dust extinction values and dust number, surface, and volume concentrations in large detail. The study presented here was also motivated by the growing PollyNET (POrtabLe Lidar sYstem NETwork) activities (Baars et al., 2016; Engelmann et al., 2016). Meanwhile, long-term observations are available, e.g., for Greece and Cyprus, Israel and United Arab Emirates, Tajikistan and South Korea, and recently also for southern Chile. Many cruises across the Atlantic from northern Germany to South Africa or South America with a Polly aboard the Research Vessel Polarstern have been conducted

in addition (Kanitz et al., 2013; Bohlmann et al., 2018). **However, as mentioned the POLIPHON technique can be applied to any available polarization lidar observation around the world.**

The paper is organized as follows. A brief overview of the POLIPHON methodology is given in Sect. 2. with focus on mineral dust and the determination of dust conversion factors from worldwide AERONET observations. We analyzed long-term sun/sky photometer observations of 20 AERONET sites in or close to important mineral dust source regions around the world (AERONET, 2019). **The stations are shown in Fig. 1**. The results (conversion factors) of the in-depth AERONET data analysis are presented in Sect. 3. In Sect. 4, we discuss a Polly observation at Dushanbe, Tajikistan, with mineral dust up to the tropopause advected from Central Asia (at heights below 2 km above ground), from Iran and the Arabia peninsula (2–7 km height range), and the Sahara (above about 8 km height). The case study is used to demonstrate the full potential of the POLIPHON method for mineral dust profiling with the updated AERONET-based dust conversion factors and also how to estimate the conversion uncertainties in the POLIPHON products. Concluding remarks are given in Sect. 5.

## 2  Methodological background

### 2.1  Summary of the POLIPHON method with focus on dust

The POLIPHON method is described in detail by Mamouri and Ansmann (2014, 2015, 2016, 2017) and with respect to the INP concentration retrieval also by Marinou et al. (2018). The main part of the POLIPHON data analysis deals with the conversion of aerosol-type-dependent particle extinction coefficients into respective particle microphysical properties. Table 1 provides an overview of POLIPHON dust products and the respective conversions. Similar conversions for non-dust aerosols such as maritime particles or continental fine-mode aerosol pollution (urban haze, biomass burning smoke) can be found in Mamouri and Ansmann (2016, 2017).

**In the first part of the POLIPHON data analysis, the polarization lidar observations are analyzed to obtain height profiles of dust and non-dust backscatter coefficients. Here we assume that pure dust causes particle linear depolarization ratios of 0.3-0.35 around the globe, disregarding the dust source region. This is corroborated by numerous studies (see the reviews in Tesche et al. (2009); Mamouri and Ansmann (2014, 2017)) and also during recent field campaigns (Groß et al., 2015; Veselovskii et al., 2016; Haarig et al., 2017; Hofer et al., 2017). More details to the aerosol type separation procedure (including the separation of fine and coarse dust by the use of fine-mode and coarse-mode-related depolarization ratios) can be found in Mamouri and Ansmann (2017). The derived total, fine, and coarse dust backscatter coefficients $\beta_\mathrm{d}$, $\beta_\mathrm{df}$, and $\beta_\mathrm{dc}$ are then converted to respective dust extinction values $\sigma_\mathrm{d}$, $\sigma_\mathrm{df}$, and $\sigma_\mathrm{dc}$ by means of appropriate dust extinction-to-backscatter ratios or lidar ratios $S_\mathrm{d}$, $S_\mathrm{df}$, and $S_\mathrm{dc}$. As shown in Table 2, the 532 nm dust lidar ratio $S_\mathrm{d}$ may vary from about 30 to 60 sr for different mineral dust types (Müller et al., 2007; Tesche et al., 2011; Mamouri et al., 2013; Nisantzi et al., 2015; Groß et al., 2015; Veselovskii et al., 2016; Haarig et al., 2017; Hofer et al., 2017; Shin et al., 2018). However, for most dust regions, except the western Sahara, the typical dust lidar ratio is 40 sr at 532 nm. We therefore recommend to use 40 sr as dust lidar ratio and to select 50 sr only in cases with airflow from the western Sahara. The best option is however to use actual Raman lidar observations of the dust lidar ratio. We further**

assume that $S_\mathrm{d} = S_\mathrm{df} = S_\mathrm{dc}$ (see lines 2-4 in Table 1). A relative uncertainty in the dust lidar ratio assumptions of 10% is considered in the estimation of the relative uncertainties (error propagation) in Table 1.

In the second part of the POLIPHON data analysis (see Table 1, lines 5-15), the height profile of the dust mass concentration $M_\mathrm{d}(z)$ is derived from the dust extinction coefficients $\sigma_\mathrm{d}(z)$, also separately for coarse dust ($M_\mathrm{dc}$ considering particles with

radius>500 nm) and fine dust ($M_\mathrm{df}$ considering dust particles with radius<500 nm) from respective coarse and fine dust extinction coefficients $\sigma_\mathrm{dc}$ and $\sigma_\mathrm{df}$. The dust extinction coefficients are converted into dust particle volume concentrations $v_d$, $v_{dc}$, and $v_{df}$ by means of extinction-to-volume conversion factors $c_\mathrm{v,d}$, $c_\mathrm{v,df}$, and $c_\mathrm{v,dc}$, and afterwards multiplied by the dust particle density $\rho_\mathrm{d}$ of $2.6\,\mathrm{g/cm}^{-3}$ (Ansmann et al., 2012) to obtain the respective dust mass concentrations. The required conversion factors are determined from AERONET observations as described in Sects. 2.2 and 3.1.

Further POLIPHON conversion products listed in Table 1 (lines 8-15) are needed in the estimation of the cloud-relevant aerosol parameters such as the cloud condensation nucleus concentration (CCNC) and ice-nucleating particle concentration (INPC). The number concentrations $n_{100,\mathrm{d}}$ (considering particles with radius >100 nm) is a good proxy for the dust CCNC (Mamouri and Ansmann, 2016; Lv et al., 2018). However, CCNC depends on the water supersaturation at cloud base where aerosol particles mainly enter the cloud and serve as CCN. A typical water supersaturation value is 0.2% (Siebert and Shaw,

2017) and occurs when air parcels are lifted into the base of a liquid water cloud by weak updrafts, e.g., in the case of fair weather cumuli. Water supersaturation values may exceed even 1% in strong updrafts. For the conversion of $\sigma_\mathrm{d}$ into number concentration $n_{100,\mathrm{d}}$, the conversion parameters $c_{100,\mathrm{d}}$ and exponent $x_\mathrm{d}$ as shown in Table 1 are used and obtained from the AERONET observations (see Sects. 2.2 anbd 3.2).

**We introduce the factor $f_{ss,\mathrm{d}}$ to consider the water supersaturation dependence. With increasing supersaturation at**

**cloud base an increasing number of dust particles (i.e., particles with lower radius) can be activated as CCN. For a supersaturation value of 0.4% even dust particles with radius of 70-80 nm become activated. According to Shinozuka et al. (2015), $f_{ss,\mathrm{d}} = 2$ is appropriate when using $n_{100,\mathrm{d}}$ as the basic aerosol parameter in the CCNC estimation but the supersaturation is 0.4% (see Mamouri and Ansmann (2016) for more details). For completeness, in Table 1, $f_{ss,\mathrm{d}}$ is 1.0, and the respective equation holds for a liquid-water supersaturation level of 0.2%.**

**The particle number concentration $n_{250,\mathrm{d}}$ (considering particles with radius >250 nm, Table 1, line 9) and the dust particle surface area concentration $s_\mathrm{d}$ (line 10) and $s_{100,\mathrm{d}}$ (considering only particles with radius >100 nm, line 11) are input in the estimation of height profiles of dust INPC when using the INPC parameterization for immersion freezing of DeMott et al. (2015) (Table 1, line 13, D15) and of Ullrich et al. (2017) (line 14, U17-I) and for deposition nucleation of Ullrich et al. (2017) (line 15, U17-D). For the conversion of $\sigma_\mathrm{d}$ into number concentration $n_{\mathbf{250,d}}$ and surface area**

**concentrations $s_\mathrm{d}$ and $s_{100,\mathrm{d}}$ the conversion factors $c_{250,\mathrm{d}}$, $c_{\mathrm{s,d}}$, and $c_{\mathrm{s,100,d}}$ are required. Besides aerosol number and surface-concentrations, the temperature profile $T(z)$ and an assumed ice supersaturation value $S_\mathrm{ice}$ (in the case of deposition-freezing INPC, U17-D) are input in the INPC estimation. The ice supersaturation $S_\mathrm{ice}$ is set to a typical value of 1.15.**

**We introduce a new parameter (not considered in Mamouri and Ansmann (2016)), namely the surface area $s_{100,\mathrm{d}}$ as**

**an alternative input parameter in the estimation of immersion freezing INPC. In the case of immersion freezing, liquid**

droplets form first before freezing occurs. As discussed above, appropriate dust CCN for typical water supersaturation values of 0.2% have a radius >100 nm. Only these particles (immersed in the liquid droplets) can then serve as INP so that the surface area $s_{100,d}$ may be a more appropriate aerosol proxy in the INP estimation by using the immersion freezing parameterization U17-I (Ullrich et al., 2017) than the total surface area concentration $s_d$. However, both pa-
rameters ($s_d$, $s_{100,d}$) are required in the INP parameterization. For deposition nucleation (heterogeneous ice nucleation by water vapor deposition directly on dust particles, without any liquid phase formation), $s_d$ is the relevant aerosol input parameter. All this is described in detail in Mamouri and Ansmann (2016). More details to the INPC retrieval are also given in Sect. 4.

     Table 1 also provides an overview of the uncertainties in the POLIPHON products (Mamouri and Ansmann, 2016,
2017). The very large uncertainties in the estimation of $n_{100,d}$, $n_{CCN}$, and $n_{INP,d}$ (factor of 2-5) are obtained when taking all potential error sources into consideration. INPC parameterizations developed from field observations (for sometimes not well characterized aerosol types) and from laboratory experiments with fresh dust particles rather than aged, i.e., chemically and cloud processed dust particles (as they frequently occur in the atmosphere), must always we be handled with care and may not be fully applicable to atmospheric conditions with predominantly aged dust so
that uncertainties of the order of a magnitude can not be excluded. However, meanwhile a variety of studies indicate that uncertainties in the $n_{CCN}$ and $n_{INP,d}$ values of the order of 50% are more realistic to characterize the errors in lidar-based CCNC and INPC estimations (Düsing et al., 2018; Marinou et al., 2018; Haarig et al., 2019). Uncertainties of 50% are acceptable in process studies of aerosol-cloud interaction performed to investigate the role of dust in cloud evolution processes. Even uncertainties of a factor of 2-5 are acceptable in attempts to establish a vertically resolved
tropospheric climatology for CCNC and INPC. Upper tropospheric long-term observations of INPC are not available in the literature, but strongly required (even if the uncertainties are high) to support weather and climate modeling.

## 2.2   POLIPHON dust conversion parameters

Trustworthy and climatologically robust conversion parameters obtained from AERONET observations are of central importance for the applicability and attractiveness of the POLIPHON method. For our study, we downloaded the
following data sets of AERONET products (single measurements, inversion products, version 3, level 2.0) (AERONET, 2019): 1) The particle volume size distribution resolved in 22 size classes from 50 nm (bin 1) to 15 nm (bin 22), 2) the corresponding data sets of total, fine-mode, and coarse-mode-related volume concentrations and effective radii (from which also surface area concentrations can be calculated), and 3) the corresponding AOTs for 8 wavelengths (denoted as extinction AOT in the AERONET data base) together with respective Ångström exponents AE for the 440-870 nm
wavelength range. Details to the AERONET data processing steps are given in Mamouri and Ansmann (2014, 2015, 2016, 2017).

     To obtain climatologically representative dust conversion factors for a given AERONET station, we filtered out all AERONET data sets fulfilling the constraints of an Ångström exponent AE<0.3 and a 532 nm AOT>0.1. The AOT for 532 nm (in the following equations simply denoted as $\tau_d$) is obtained from the 500 nm AOT $\tau_{500}$ and the Ångström

**exponent $a$, stored in the AERONET data base, by**

$$\tau_d = \tau_{500}(500/532)^a. \tag{1}$$

**More information to the dust selection criteria are given in Sect. 3.**

It is noteworthy to mention that recent airborne in situ observations of dust size distributions over the Sahara and re-
mote dust outflow regions by Ryder et al. (2019) corroborate the high quality and consistence of the overall AERONET
optical and microphysical data sets and the applicability of the AERONET data analysis and inversion concept. The
AERONET inversion method required to obtain the aerosol microphysical from the measured optical properties as-
sumes that only particles with radius $\leq$15 $\mu$m are responsible for the observed dust-related optical effects. The presence
of larger dust particle is ignored. Ryder et al. (2019) now show that dust particles with radius $>$15 $\mu$m contribute by
only 1-3% to the particle extinction coefficient at 550 nm. This means that this size cutoff effect has practically no
impact on the AERONET inversion products and thus on the derived POLIPHON conversion factors.

In the following, we use the example of the dust mass concentration retrieval to explain the basic idea of the derivation of
conversion factors from the AERONET data base. The mass concentration for the aerosol type dust (index d) is given by

$$M_d(z) = \rho_\mathrm{d} \times v_\mathrm{d}(z) \tag{2}$$

with the dust particle density $\rho_\mathrm{d}$ of 2.6 g cm$^{-3}$ and the dust volume concentration $v_\mathrm{d}$. The required dust volume concentration
in Eq. (2) can be obtained from the following conversion:

$$v_\mathrm{d}(z) = c_{\mathrm{v,d},\lambda} \times \sigma_{\mathrm{d},\lambda}(z) \tag{3}$$

with the extinction-to-volume conversion factor $c_{\mathrm{v,d},\lambda}$ (derived from the AERONET long term observations) and the particle
extinction coefficient $\sigma_{\mathrm{d},\lambda}$ measured with lidar at wavelength $\lambda$. We concentrate on lidar observations at 532 nm in this study
and omit the wavelength index $\lambda$ in the following. The conversion factor is obtained from the AERONET observations of the
vertically integrated particle volume concentration $V_\mathrm{d}$ (denoted also as column volume concentration) and the aerosol optical
thickness $\tau_\mathrm{d}$ (AOT at 532 nm, see Eq. 1),

$$c_{\mathrm{v,d}} = \frac{V_\mathrm{d}}{\tau_\mathrm{d}}. \tag{4}$$

To provide a link to the lidar-derived height profile of $\sigma_\mathrm{d}(z)$ (see Eq. 3), we introduce an aerosol layer depth with an
arbitrarily chosen vertical extent $D$. With $D$, Eq. (4) can be written as

$$c_{\mathrm{v,d}} = \frac{V_\mathrm{d}/D}{\tau_d/D} = \frac{v_\mathrm{d}}{\sigma_\mathrm{d}} \tag{5}$$

with the layer mean volume concentration $v_\mathrm{d}$ and the layer mean particle extinction coefficient $\sigma_\mathrm{d}$. For simplicity, we assume
that all aerosol is confined to the introduced layer with vertical depth $D$. We may interpret this layer as the dust containing
boundary layer or as a lofted dust layer with a vertical extent $D$. The introduced layer depth $D$ has no impact on the further

retrieval of the conversion factors and is only required to move from column-integrated values and AOT to more lidar-relevant quantities like concentrations and extinction coefficients.

To obtain climatologically representative dust conversion factor for a given AERONET station, we selected all individual dust observations (from number $j = 1$ to $J_\mathrm{d}$ collected over many years), as mentioned defined by an Ångström exponent AE<0.3 and 532 nm AOT>0.1. For each dust observation $j$ we computed $c_{\mathrm{v,d},j}$ and then determined the mean value, which we interpret as the climatologically representative POLIPHON conversion factor,

$$c_\mathrm{v,d} = \frac{1}{J_\mathrm{d}} \sum_{j=1}^{J_\mathrm{d}} \frac{v_{\mathrm{d},j}}{\sigma_{\mathrm{d},j}}. \tag{6}$$

In the same way, all other conversion parameters in Table 1 are computed:

$$c_\mathrm{v,dc} = \frac{1}{J_\mathrm{d}} \sum_{j=1}^{J_\mathrm{d}} \frac{v_{\mathrm{dc},j}}{\sigma_{\mathrm{d},j}}, \tag{7}$$

$$c_\mathrm{v,df} = \frac{1}{J_\mathrm{d}} \sum_{j=1}^{J_\mathrm{d}} \frac{v_{\mathrm{df},j}}{\sigma_{\mathrm{d},j}}, \tag{8}$$

$$c_\mathrm{250,d} = \frac{1}{J_\mathrm{d}} \sum_{j=1}^{J_\mathrm{d}} \frac{n_{\mathrm{250,d},j}}{\sigma_{\mathrm{d},j}}, \tag{9}$$

$$c_\mathrm{s,d} = \frac{1}{J_\mathrm{d}} \sum_{j=1}^{J_\mathrm{d}} \frac{s_{\mathrm{d},j}}{\sigma_{\mathrm{d},j}}, \tag{10}$$

$$c_\mathrm{s,100,d} = \frac{1}{J_\mathrm{d}} \sum_{j=1}^{J_\mathrm{d}} \frac{s_{\mathrm{100,d},j}}{\sigma_{\mathrm{d},j}}. \tag{11}$$

As before, indices df and dc denote fine-mode and coarse-mode dust fractions, respectively. In Mamouri and Ansmann (2015, 2016), we explain how we calculate $n_{\mathrm{250,d},j}$, $s_{\mathrm{d},j}$ as well as $s_{\mathrm{100,d},j}$ (discussed below) from the downloaded AERONET size distribution data sets.

In the retrieval of the conversion parameters required to obtain $n_\mathrm{100,d}$ (Table 1, line 8), we used a different approach. Following the procedure suggested by Shinozuka et al. (2015), we applied a log-log regression analysis to the $\log(n_\mathrm{100,d})$-$\log(\sigma_\mathrm{d})$ data field for each of the considered 20 AERONET station and determined in this way representative values for $c_\mathrm{100,d}$ and $x_\mathrm{d}$ that fulfill best the relationship,

$$\log(n_\mathrm{100,d}) = \log(c_\mathrm{100,d}) + x_\mathrm{d} \log(\sigma_\mathrm{d}), \tag{12}$$

as will be shown in the next section.

## 3  Conversion parameters from the AERONET data base

Table 3 contains the list of AERONET stations considered in our effort to determine dust conversion factors for different desert regions around the globe. Measurement periods, numbers of available individual observations and of dust-related observations

($J_d$) in Eqs. (6)-(11), and mean aerosol and dust conditions are given as well in Table 3. The locations of the AERONET sites are shown in Fig. 1. We preferred stations in Africa, Middle East and Asia with long data records and large numbers of observations. As can be seen in Table 3, the number of useful dust observations (AE<0.3, AOT>0.1) ranges from 218–4199 for 13 out of the 20 sites and is thus sufficiently high enough for the statistical analysis. The first six stations (from Tamanrasset to Ilorin) in Table 3 are exclusively influenced by Saharan dust, the next six stations (Limassol to Mezaira) by Saharan and Middle East (mainly Arabian desert) dust, followed by three stations in Central and East Asia (Dushanbe to Dalanzadgad), which are influenced by long-range transport from the Sahara and western Asian deserts (including deserts in Iran and Kazakhstan) but also strongly by desert dust from Taklamakan and Gobbi deserts (Langzhou, Dalanzadgad). The Limassol data sets belongs to the Sahara group because the majority of dust outbreaks contain Saharan dust (Nisantzi et al., 2015). Sun photometer observations in North America (Great Basin, Tuscon, White-Sands), South America (Patagonian desert, Trelew), South Africa (Kalahari desert, Gobabeb) and in the central Australian desert (Birdsville) complete our global AERONET dust data set. It was difficult to find AERONET stations in North and South Amercia with a useful number of cases indicating pure dust observations.

We defined an ambitious, quite demanding criterion to filter out the pure dust cases for our study. The constraints AOT>0.1 at 532 nm and Ångström exponents AE<0.3 for the 440-870 nm wavelength range guarantee that interference by anthropogenic pollution, biomass burning smoke, and marine particles are of minor importance. The mean values and standard deviations for dust AOT in Table 3 indicate that even for stations with relatively low mean dust AOT (Izana in the free troposphere, and for the stations in Amercia and Australia), the impact of marine aerosol (showing approximately the same size distribution and AE characteristics as mineral dust and causing AOT of around 0.05) was still low. A sensitive impact of fine-mode-dominated fire smoke and urban haze on the conversion calculations is also unlikely as long as AE<0.3. Additional smoke contributions immediately lead to AE values clearly above 0.5 as our field campaign experience indicate (Tesche et al., 2009; Nisantzi et al., 2014; Hofer et al., 2017).

## 3.1 Correlations between $n_{250,d}$, $s_d$, $s_{100,d}$ and $v_d$ with dust extinction coefficient $\sigma_d$

In order to illustrate the variability in the POLIPHON conversion factors (summarized in Sect. 3.3 and Table 4), we start with basic correlations between the dust microphysical properties and the dust extinction coefficient. Figure 2 provides an overview of the relationship between the dust particle number concentration of larger particles $n_{250,d}$ and the dust extinction coefficient $\sigma_d$ in (a), dust particle surface area concentration $s_d$ and $\sigma_d$ in (b), and between the dust volume concentration $v_d$ and $\sigma_d$ in (c). Twelve different AERONET stations are considered in the figure. The mean conversion factors $c_{250,d}$ (Eq. 9), $c_{s,d}$ (Eq. 10), and $c_{v,d}$ (Eq. 6) are indicated as straight lines (regression lines) for the Saharan dust stations of Sal, Cabo Verde (a), Dakar, Senegal (b), and Tamanrasset, Algeria (c). These stations are exclusively influenced by Saharan dust.

We set the layer depth $D$ in Eq. (5) simply to 1000 m so that $\sigma_d$ (in Mm$^{-1}$ in Fig. 2) divided by 1000 yield the basic AERONET 532 nm AOT value. We selected different colors to distinguish Saharan dust observations (green), Middle East measurements (orange) and data collected in Central and East Asia (red). We used bluish colors (blue, cyan) for the American and Australian stations, respectively, and blue-green for the African site (in the southern hemisphere) of Gobabeb.

As can be seen in Fig. 2, there are no large differences in the correlation features for the different AERONET stations. The given Saharan dust conversion factors in (a) for Cabo Verde (based on 2982 data points), in (b) for Dakar, Senegal (3823 data points), and in (c) for Tamanrasset, Algeria (3542 data points) characterize very well the main relationship between the shown microphysical and optical parameters for the different dust regions. However, some contrasting features are visible, especially when comparing the Saharan with the Central and East Asian station and thus for clearly separated dust regions. The Middle East AERONET sites are influenced by both Saharan as well as Middle East (mainly Arabian) dust.

The spread in the data mainly reflects variations in the dust aerosol characteristics (size distribution, refractive index) as a function of varying mixtures of freshly emitted local dust and long-range-transported aged dust. Fresh and aged dust mixtures may have occurred in different dust layers above each other (as in the case study in Sect. 4). Uncertainties in the AERONET data inversion procedure applied to obtain the microphysical properties from the measured AOT and sky radiance observations may have also contributed to the scatter in the data. The scatter provides an impression of the variability in the relationship between dust microphysical and optical properties and thus indicates the uncertainty in the determined conversion factors. However, it should also be mentioned that dust extinction coefficients in lofted layers above the boundary layer (in the free troposphere) seldom exceed 200-300 $\mathrm{Mm}^{-1}$. For $\sigma_\mathrm{d} <$500 $\mathrm{Mm}^{-1}$ the scatter in the data is comparably low in Fig. 2.

**Figure 3 indicates that the relationship between the surface area concentration $s_{100,\mathrm{d}}$, i.e., the CCN-related particle surface concentration, and the particle extinction coefficient $\sigma_\mathrm{d}$ at 532 nm is much more robust (less variable) than the one for $s_\mathrm{d}$ vs $\sigma_\mathrm{d}$ in Fig. 2b. The reason for this less noisy relationship is probably that the AERONET inversion analysis (for coarse-mode dominated particle ensembles) is not very accurate for the small-particle fraction (radius classes from 50-100 nm) and this inversion-related uncertainty is then reflected in the variability of the $s_\mathrm{d}$ values considering all particle classes. With increasing minimum particle radius in the surface area computation the variability in the relationship between respective surface area concentration and extinction coefficient decreases.**

**However, as will be shown in the next section, in contrast to the $s_{100,\mathrm{d}}$ vs $\sigma_\mathrm{d}$ relationship, the correlation between $n_{100,\mathrm{d}}$ and $\sigma_\mathrm{d}$ is strongly variable. One of the reason for this difference is that particles with large geometrical cross section (coarse-mode particles) have a higher weight in the surface area computation (integral over all sizes classes) and thus control the $s_{100,\mathrm{d}}$ values. In the $n_{100,\mathrm{d}}$ calculation, on the other hand, the size classes with highest particle number concentration (fine-mode classes) dominate the $n_{100,\mathrm{d}}$ values.**

### 3.2 Relationship between $n_{\mathbf{100},d}$ and dust extinction coefficient $\sigma_\mathbf{d}$

A different way of data analysis is used for $n_{100,\mathrm{d}}$. As suggested by Shinozuka et al. (2015) we correlated $\log(n_{100,\mathrm{d}})$ vs $\log(\sigma_\mathrm{d})$. Figure 4 shows the relationship between particle number concentration $n_{100,\mathrm{d}}$ and the dust extinction coefficient $\sigma_\mathrm{d}$ at 532 nm for two stations (Mezaira, Dushanbe) in logarithmic scale. As outlined in Sect. 2, the particle number concentration $n_{100,\mathrm{d}}$, considering only the particles with dry radius $>$100 nm, represents very well the CCN reservoir in the case of dust particles for a typical water supersaturation of 0.2% (Mamouri and Ansmann, 2016; Lv et al., 2018).

In Fig. 4, we highlight the difference in the correlation when using all available data (532 nm dust AOT from 0.1 to 3.0 or $\sigma_\mathrm{d}$ from 100-3000 $\mathrm{Mm}^{-1}$) and when using only observations with AOT$<$0.6. By detailed inspection of all data sets (station by

station), we observed that the correlation strength significantly decreases with increasing AOT and is no longer clearly visible for all measurements with AOT from 1.0 to 3.0. The Dushanbe data set shown in Fig. 4b is a good example for this observation.

We can only speculate about the reason for the weak relationship for AOT>0.6. When the AOT is too large, the coarse-mode dust fraction may control the measured optical properties and respective inversion results so much that a trustworthy retrieval of the particle fraction with radii from, e.g., 100–200 nm is no longer possible. Another explanation is related to the observational procedure. Most inversion computations are based on AERONET observations in the early morning and evening hours when the effective impact of aerosols is strongest (so that the effective dust AOT is even higher by a factor of two and more than the one for the vertical column stored in the AERONET data base). At these low-visibility conditions, the short-wavelength AERONET channels (340 and 380 nm) may have problems to correctly measure the overall AOT (Rayleigh AOT plus particle AOT). The short-wavelength AOT values are, however, especially important in the inversion retrieval of small dust particles and thus have a strong influence on the $n_{100,\mathrm{d}}$ retrieval results.

As a consequence of the low correlation between $\log(n_{100,\mathrm{d}})$ and $\log(\sigma_\mathrm{d})$ for large AOT we restricted the determination of the conversion parameters $c_{100,\mathrm{d}}$ and $x_\mathrm{d}$ (see Eq. 12) by means of a regression analysis to AOT values from 0.1-0.6 (or respective $\sigma_\mathrm{d}$ from 100-600 Mm$^{-1}$).

Figure. 5 provides further insight into the correlation between $\log(n_{100,\mathrm{d}})$ and $\log(\sigma_\mathrm{d})$. Observations for different stations influenced by Saharan, Middle East, central Asian, American, and Australian dust are shown. The Saharan dust data set collected at Cabo Verde belongs to the few data sets (out of the 20 AERONET stations) with a likewise good correlation between $\log(n_{100,\mathrm{d}})$ and $\log(\sigma_\mathrm{d})$ even for large extinction values >600 Mm$^{-1}$ and corresponding AOT values >0.6. In Fig. 5, regression analysis results (in accordance with Eq. 12) for Mezaira (numbers in orange) and Cabo Verde (numbers in green) are compared. Furthermore, the relationship between $n_{100,\mathrm{d}}$ and $\sigma_\mathrm{d}$ as found by Shinozuka et al. (2015) for dusty field sites is presented.

As mentioned above, most of the dust-related lidar observations in the free troposphere show dust extinction coefficients ($\sigma_\mathrm{d}$) <200-300 M$^{-1}$. For a moderate dust extinction value of 100 Mm$^{-1}$, the POLIPHON retrieval yields $n_{100,\mathrm{d}} \approx 150$ cm$^{-3}$ and 250 cm$^{-3}$ when using $c_{100,\mathrm{d}}$ and $x_\mathrm{d}$ numbers as derived from the Cabo Verde and Mezaira AERONET observations (AOT<0.6), respectively. Thus, a maximum overall error of a factor 2 in Table 1 (for $n_{100,\mathrm{d}}$ and $n_{\mathrm{CCN,d}}$) also concluded by Shinozuka et al. (2015) and corroborated by Mamouri and Ansmann (2016) is justified.

### 3.3 Overview of AERONET-derived conversion parameters

In Table 4, the AERONET-based conversion parameters for all stations are presented. Regional mean sets of conversion parameters are given as well. Figures 6 and 7 provide a station-by-station overview of the conversion parameters (mean and SD values). Systematic differences from region to region are visible in the case of $c_{250,\mathrm{d}}$ and also weakly for $c_{\mathrm{v,d}}$. The conversions parameters for the American, Australia, and southern Africa need to be handled with caution because the number of available observations is relatively low and the mean 532 nm AOT of these observations was low as well with values from 0.15–0.25. A decrease in $c_{250,\mathrm{d}}$ and a slight increase in $c_{\mathrm{v,d}}$ (and $c_{\mathrm{v,dc}}$) from African to East Asian AERONET stations suggests that, for the same measured extinction coefficient ($\sigma_\mathrm{d}$), the accumulation mode particle number concentration (in our case particles with radius from 250–500 nm) is slightly larger and the coarse mode dust particle number concentration, dominating the

dust volume concentration, is lower in the case of Saharan dust compared to East Asian dust. This behavior may indicate that the African AERONET stations, e.g., in Cabo Verde, Izana, and Dakar observe predominantly dust after long-range transport (which leads to a bit enhanced fine dust fraction because of size dependent sedimentation and removal of particles), whereas the East Asian AERONET stations may be influenced more frequently by the occurrence of local, freshly emitted dust with the relatively strong contribution of coarse-mode particles. Similar conditions as suggested for Central and East Asia may hold for the American and Australian stations.

The smooth but steady changes in $c_{250,\mathrm{d}}$ and $c_{\mathrm{v,d}}$ from the Saharan, over the Middle East to the central and eastern Asian AERONET stations indicates that the Middle East stations are influenced by both, local, western Asian dust sources (mostly Arabian dust) and Saharan dust (with the prevailing westerly winds). Only the African stations and the East Asian stations at Lanzhou and Dalanzadgad are clearly separated and allow to contrast dust properties of African and Asian deserts. **We did not make an attempt to separate Saharan from Arabian dust observations in case of the Middle East data set by using backward trajectory analysis because of the relatively small differences in the Saharan and Middle East conversion factors and the likewise large variability bars.** All in all, the observed regional differences in the dust conversion parameters in Fig. 6 are of the order of $\pm$15-20% for most of the parameters and stations.

Fig. 7 provides a summarizing overview of the final results for $c_{100,\mathrm{d}}$ and $x_{\mathrm{d}}$. Because of the large scatter in the log-log data fields expressed in the large uncertainty bars in the figure we can only give recommendations regarding the selection of the most reasonable set of dust conversion parameters. For the extinction exponent $x_{\mathrm{d}}$ a value of 0.80 seems to be appropriate. This exponent is then linked to $c_{100,\mathrm{d}}$ values of 5–6 cm$^{-3}$ (at $\sigma_{\mathrm{d}}$ =1 Mm$^{-1}$).

## 4   Lidar measurement example: Case study of a dust observation in Tajikistan

We used the updated set of conversion parameters to analyze a dust measurement performed with a Polly system deployed at Dushanbe (38.6°N, 68.9°E, 820 m a.s.l.), Tajikistan, in April2015. The lidar observations were performed in the framework of an 18-month field campaign CADEX (Central Asian Dust Experiment) (Hofer et al., 2017). The full potential of the POLIPHON method (Table 1) is shown. In addition, the impact of the selected conversion factors on the results is illuminated in the framework of an uncertainty analysis. The case presented here was already discussed in terms of optical properties by Hofer et al. (2017).

Figure 8 presents an overview of the aerosol conditions observed with lidar on 13 April 2015. A pronounced dust layer was detected between 2 and 5 km height (above ground level, AGL, about 3-6 km height above sea level, a.s.l.). Dust was observed up to cirrus heights. The AERONET sun photometer observations at Dushanbe showed a 500 nm AOT of 0.4, AE of 0.2, and a fine-mode fraction (FMF) of 0.2 (just before sunset close to 13:00 UTC). Thus, fine dust contributed about 20% to the overall (fine and coarse) dust extinction coefficient. According to the backward trajectories in Fig. 9, mineral dust in the polluted boundary layer (0-2 km height) originated from Kazakhstan and local dust sources. The dust particles in the thick dust layer from to 2–5 km height were mostly emitted in Iran and Oman. Higher up (above 5 km) long range transport of dust from the

Arabian peninsula (5-7 km height) and even the Sahara (8-10 km) prevailed. More details to the long-range transport features in comparison with aerosol transport modeling is given in Hofer et al. (2017).

Figure 10a shows the basic lidar profiles used in the POLIPHON data analysis. The height profiles of the particle (dust + non-dust) backscatter coefficient and the related particle linear depolarization ratio are used to derive the dust and non-dust extinction profiles (Mamouri and Ansmann, 2014, 2017). The dust extinction coefficients are then converted into the dust mass concentrations in Fig. 10b by means the dust conversion factor $c_{v,d}$ in Table 4 for Dushanbe (red profiles). The mass computation is performed in the way described in Table 1. The corresponding dust mass fraction (ratio of dust mass concentration to total particle mass concentration) is presented in Fig. 10b as well. To provide an estimate of the uncertainty in the dust mass concentration introduced by the conversion uncertainty two conversion factors for Dushanbe and for Cabo Verde, representing a relatively high and low value of all conversion factors listed in Table 4, were applied in Fig. 10. The resulting differences in the POLIPHON results are well covered by the overall uncertainty in the POLIPHON mass retrieval of 30% (see the error bars in Fig. 10) which also includes the uncertainty in the dust extinction determination.

Figure 11 presents the POLIPHON results in terms of several CCNC profiles obtained with conversion parameter sets for Cabo Verde, Mezaira, and Dushanbe (see Table 4). As mentioned, $n_{CCN,d} \approx n_{100,d}$ for a water supersaturation value of 0.2%. According to the discussion in Sect. 3.2 and the uncertainty information in Table 1 the overall uncertainty in the regression analysis of $n_{100,d}$ with $\sigma_d$ is of the order of 50-200%. In Fig. 11, an uncertainty factor of 2 is considered by the dashed lines. Compared to this factor-2 uncertainty margin, the impact of the applied different conversion parameter sets is likewise small.

Figure 12 shows the POLIPHON results in terms of ice-nucleating particle concentrations. As outlined in detail in Mamouri and Ansmann (2016), the POLIPHON data analysis delivers height profiles of the large-particle number concentration $n_{250,d}$ and of the dust surface area concentration $s_d$ in Fig. 12a with an accuracy of about 25-30% in the case of pronounced dust layers. Again, we applied two contrasting conversion parameter sets (Dushanbe, Cabo Verde). The differences in the results are well covered by the overall POLIPHON uncertainties of 30%.

The profile of $n_{250,d}$ is then input in the INPC computation by means of the immersion-freezing parameterization of DeMott et al. (2015) (see Fig. 12b, D15 profile) and the profile of $s_d$ is needed as input in the INPC parameterization of Ullrich et al. (2017) (deposition nucleation mode, U17-D profile in Fig. 12b). Besides the aerosol profiles, actual GDAS temperatures (indicated as horizontal grey lines in Fig. 12b) are required in the calculations of $n_{INP,d}$ profiles. Deposition freezing usually takes place in the upper troposphere at temperatures clearly below $-30°C$ and depends on the ice supersaturation level $S_{ice}$ in an ascending air parcel. $S_{ice}$ is set to 1.15 in Fig. 12b. Dashed lines indicate an uncertainty of a factor of 3 (one order of magnitude) caused by the INPC parameterization schemes. Compared to this uncertainty the impact of an uncertainty in the conversion factors on the relative error of the $n_{250,d}$ and $s_d$ values and finally on the accuracy of the INPC estimates is of minor importance.

In Fig. 12b, we added an INPC profile segment (from 8.5–10 km) based on observations in cloud free air from 15:15-16:10 UTC (see Fig. 8, just before the time period indicated by the white frame) to extend the INPC profile up to the height range where several cirrus layers formed. The patchy ice cloud cluster at 7 km between 15:00 and 16:30 UTC (see Fig. 8) probably formed via immersion freezing at temperatures around $-25°C$. INPC was high with 1-10 $L^{-1}$, and thus triggered the

nucleation of 1-10 ice crystals per liter. Higher up, at 9-11 km and corresponding temperatures from $-35$ to $-50°$C, deposition freezing prevails besides homogeneous freezing. The use of the U17-D parameterization indicates INPC values of 0.1-1 $L^{-1}$ which is relatively low and may explain the short-lived thin ice cloud features (occurring after 16:15 UTC) and the absence of large cirrus fields with extended virga zones.

## 5   Conclusions

An extended global AERONET analysis has been performed to create a global data set of dust-related POLIPHON conversion factors. We analyzed AERONET observations for all relevant desert regions in Africa, Middle East, Central and East Asia, America, and Australia and provide respective regional conversion parameter sets. Significant differences in the obtained conversion parameters caused by potentially different dust composition and size distribution characteristics for different desert regions were not found. Furthermore, the presented Tajikistan case study showed that the use of different, contrasting conversion parameters did not have large (dominating) impact on the overall uncertainty in the POLIPHON results. **This is of advantage for spaceborne lidar applications when one wants to use, e.g., one set of conversion parameters in global observations. This universal conversion parameter set may be the mean of all Saharan, Middle East, and Asian dust conversion parameters given in Table 4. For ground-based observations it is however always advisable to make use of the specific, regional conversion parameters and to check the uncertainty caused by the conversion by using different conversion parameter sets listed in Table 4.**

In conclusion, we can state that appropriate conversion parameters are now available for mineral dust around the globe. In addition, conversion parameters representing pure marine conditions are available from marine Barbados AERONET observations (Mamouri and Ansmann, 2016, 2017). As an outlook, it remains to investigate in detail the conversion parameters for anthropogenic aerosol particles (urban haze, rural background aerosol, forest fire smoke, and free tropospheric smoke and haze by using mountain stations). A detailed study for anthropogenic aerosol conversion parameters has only be done so far for the urban, highly polluted AERONET stations of Leipzig and Limassol (Mamouri and Ansmann, 2016, 2017).

## 6   Data availability

All data used in this work can be accessed through the AERONET home page at https://aeronet.gsfc.nasa.gov/ (last access: 22 Februray 2019). Polly lidar observations (level 0 data, measured signals) are in the PollyNET data base (http://polly.rsd.tropos.de/). All the analysis products are available at TROPOS upon request (info@tropos.de).

## 7   Author contributions

AA and REM worked on the applied methodology and prepared the manuscript. JH provided the Dushanbe case study results. DA, JH, and SFA took care of the excellent performance of the Polly lidar and AERONET photometer during the 18-month CADEX field campaign.

# 8 Competing interests

The authors declare that they have no conflict of interest.

*Acknowledgements.* We are grateful to all PIs of the AERONET sites used in this study for maintaining their instruments and providing their data to the community. We thank AERONET for their continuous efforts in providing high-quality measurements and products. The authors acknowledge funding from the Horizon 2020 research and innovation program ACTRIS-2 Integrating Activities (H2020-INFRAIA-2014-2015, grant agreement no. 654109). We thank AERONET-Europe for providing calibration service. AERONET-Europe is also part of the ACTRIS-2 project. The CADEX project was funded by the German Federal Ministry of Education and Research in the context of 'Partnerships for sustainable problem solving in emerging and developing countries' under the grant number 01DK14014. Aerosol sources apportionment analysis has been supported by air mass transport computation with the NOAA (National Oceanic and Atmospheric Administration) HYSPLIT (HYbrid Single-Particle Lagrangian Integrated Trajectory) model using GDAS meteorological data. We also thank the three reviewers for very fruitful suggestions.

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

**Table 1.** Overview of the dust-related computations and conversions within the POLIPHON data analysis. The needed conversion factors $c_{v,d}$, $c_{v,df}$, $c_{v,dc}$, $c_{s,d}$, $c_{s,100,d}$, $c_{100,d}$, and $c_{250,d}$ are listed in Table 4. Besides the aerosol backscatter and extinction input profiles, $\beta(z)$ [Mm$^{-1}$ sr$^{-1}$] and $\sigma(z)$ [Mm$^{-1}$], the temperature profile $T(z)$ [K] is required in the $n_{INP,d}$ estimation. $r$ denotes the radius of the particles. **Uncertainties (right column) are discussed in Mamouri and Ansmann (2016, 2017). Minimum values of the given uncertainty ranges represent typical relative errors in the case of moderate to strong dust concentrations. The maximum values consider the potentially high uncertainties in the conversion factors, the needed input parameters, and applied INP parameterizations. See the text in Sect. 2 for more details of the different retrieval steps, input parameters, and products.**

| Dust parameter | Product/computation | Input profiles | Uncertainty |
|---|---|---|---|
| Backscatter coef. (total, fine, coarse)[Mm$^{-1}$ sr$^{-1}$] | $\beta_d(z), \beta_{df}(z), \beta_{dc}(z)$ | $\beta_p(z), \delta_p(z)$ | 10–30% |
| Extinction coefficient [Mm$^{-1}$] | $\sigma_d(z)=S_d\beta_d(z)$ | $\beta_d(z)$ | 15–25% |
| Fine-mode extinction coef. [Mm$^{-1}$] | $\sigma_{df(z)}=S_{df}\beta_{df}(z)$ | $\beta_{df}(z)$ | 30–50% |
| Coarse-mode extinction coef. [Mm$^{-1}$] | $\sigma_{dc}(z)=S_{dc}\beta_{dc}(z)$ | $\beta_{dc}(z)$ | 20–30% |
| Mass concentration [$\mu$g m$^{-3}$] | $M_d(z) = \rho_d c_{v,d}\sigma_d(z)$ | $\sigma_d(z)$ | 20–30% |
| Fine-mode mass conc. [$\mu$g m$^{-3}$] | $M_{df}(z) = \rho_d c_{v,df}\sigma_{df}(z)$ | $\sigma_{df}(z)$ | 40–60% |
| Coarse-mode mass conc. [$\mu$g m$^{-3}$] | $M_{dc}(z) = \rho_d c_{v,dc}\sigma_{dc}(z)$ | $\sigma_{dc}(z)$ | 25–35% |
| Particle number conc. ($r >$100 nm) [cm$^{-3}$] | $n_{100,d}(z) = c_{100,d} \times \left(\frac{\sigma_d(z)}{1\ \mathrm{Mm}^{-1}}\right)^{x_d}$ | $\sigma_d(z)$ | 50-200% |
| Particle number conc. ($r >$250 nm) [cm$^{-3}$] | $n_{250,d}(z) = c_{250,d} \times \sigma_d(z)$ | $\sigma_d(z)$ | 25-35% |
| Particle surface conc. [m$^2$ cm$^{-3}$] | $s_d(z) = c_{s,d} \times \sigma_d(z)$ | $\sigma_d(z)$ | 30-40% |
| Particle surface conc. ($r >$100 nm) [m$^2$ cm$^{-3}$] | $s_{100,d}(z) = c_{s,100,d} \times \sigma_d(z)$ | $\sigma_d(z)$ | 20-30% |
| CCN concentration [cm$^{-3}$] | $n_{CCN,ss,d}(z) = f_{ss,d} \times n_{100,d}(z)$ | $n_{100,d}(z)$ | 50-200% |
| INP concentration [L$^{-1}$] | $n_{INP,d}(z)$ (e.g., D15) | $n_{250,d}(z), T(z)$ | 50-500% |
| INP concentration [L$^{-1}$] | $n_{INP,d}(z)$ (e.g., U17-I) | $s_{100,d}(z)$ or $s_d(z), T(z)$ | 50-500% |
| INP concentration [L$^{-1}$] | $n_{INP,d}(z)$ (e.g., U17-D) | $s_d(z), T(z)$ | 50-500% |

**Table 2.** Lidar ratios for different desert regions from AERONET observations at 675 nm (Shin et al., 2018) and from numerous lidar observations at 532 nm. Recommendations for lidar ratios to be used in the POLIPHON data analysis are given in the right column.

| Desert | AERONET [675 nm] | Lidar [532 nm] | POLIPHON [532 nm] |
|---|---|---|---|
| North Africa (Sahara, west) | 42-57 sr | 45-60 sr | 50 sr |
| North Africa (Sahara, central, east) | 42-57 sr | 40-50 sr | 40 sr |
| Middle East deserts | 33-41 sr | 35-45 sr | 40 sr |
| Asian deserts (central, Gobi) | 36-46 sr | 35-45 sr | 40 sr |
| North America (Great Basin) | 28-38 sr | – | 40 sr |
| Australia (Great Victoria) | 30-36 sr | – | 40 sr |

**Table 3.** Overview of AERONET stations considered in the study, selected observational periods for which version-3 level-2.0 data are available (Giles et al., 2019), total number of observations (inversion products), 532 nm AOT (mean and SD), dust-related inversion cases (AE<0.3, AOT>0.1), and 532 nm AOT (mean and SD) for the dust observations only.

| AERONET site | Acronym | Time period, level-2.0 data | Obs. | AOT | Dust obs. | Dust AOT |
|---|---|---|---|---|---|---|
| Tamanrasset, Algeria | TA | 30 Sep 2006 – 19 Jun 2018 | 7442 | 0.23±0.24 | 3542 | 0.37±0.28 |
| Izana, Tenerife, Spain | IZ | 1 Nov 2004 –22 May 2018 | 3264 | 0.07±0.11 | 499 | 0.26±0.14 |
| Sal, Cabo Verde | CV | 2 Nov 1994 – 9 Jun 2017 | 4718 | 0.36±0.27 | 2982 | 0.45±0.28 |
| Dakar, Senegal | DK | 24 Jun 2000 – 12 Feb 2018 | 7985 | 0.45±0.29 | 3823 | 0.60±0.33 |
| Banizoumbou, Niger | BA | 17 Oct 1995 – 15 Mar 2017 | 8547 | 0.46±0.34 | 3875 | 0.65±0.39 |
| Ilorin, Nigeria | IL | 25 Apr 1998 – 26 Mar 2018 | 4024 | 0.87±0.47 | 466 | 1.20±0.59 |
| Limassol, Cyprus | LI | 14 Apr 2010 - 5 May 2017 | 2606 | 0.17±0.11 | 72 | 0.43±0.22 |
| Eilat, Israel | EI | 26 Nov 2007–22 June 2018 | 7213 | 0.20±0.14 | 657 | 0.39±0.26 |
| Sede Boker, Israel | SB | 16 Oct 1995 – 14 Jan 2018 | 17005 | 0.17±0.28 | 1610 | 0.35±0.23 |
| Nes Ziona, Israel | NZ | 17 Dec 2000 – 14 Nov 2015 | 5268 | 0.21±0.16 | 410 | 0.48±0.32 |
| Solar Village, Saudi Arabia | SV | 23 Feb 1999 – 15 Dec 2012 | 14284 | 0.33±0.23 | 4199 | 0.51±0.30 |
| Mezaira, United Arab Emirates | ME | 8 Jun 2004 – 8 May 2018 | 7354 | 0.32±0.20 | 1055 | 0.55±0.28 |
| Dushanbe, Tajikistan | DU | 5 Jul 2010– 11 Apr 2018 | 3808 | 0.29±0.20 | 325 | 0.65±0.40 |
| Lanzhou(SACOL), China | LA | 28 Jun 2006 – 3 May 2013 | 3384 | 0.32±0.19 | 218 | 0.68±0.37 |
| Dalanzadgad, Mongolia | DA | 27 Mar 1998 – 25 Dec 2017 | 2577 | 0.10±0.09 | 49 | 0.29±0.16 |
| Tuscon, Arizona, USA | TU | 24 Nov 1993 – 18 Apr 2018 | 4881 | 0.06±0.06 | 17 | 0.15±0.04 |
| White-Sands, New Mexico, USA | WS | 17 Nov 2006 – 23 Jun 2018 | 6696 | 0.05±0.04 | 27 | 0.22±0.12 |
| Trelew, Argentina | TR | 11 Nov 2005 – 12 Oct 2017 | 2770 | 0.04±0.03 | 21 | 0.16±0.05 |
| Gobabeb, Namibia | GO | 11 Nov 2014 – 29 Jul 2018 | 5117 | 0.08±0.08 | 89 | 0.15±0.05 |
| Birdsville, Australia | BI | 13 Aug 2005 – 17 Dec 2017 | 6578 | 0.04±0.04 | 59 | 0.25±0.12 |

**Table 4.** Dust conversion parameters required in the conversion of particle extinction coefficients $\sigma_d$ at 532 nm into particle number, surface area and volume concentration (index d for total dust, index df for fine dust, index dc for coarse dust) as described in Table 1. The mean values and SD for $c_{v,d}$, $c_{v,df}$, and $c_{v,dc}$ (in $10^{-12}$ Mm), of $c_{250,d}$ (in Mm cm$^{-3}$), and $c_{s,d}$ and $c_{s,100,d}$ (in $10^{-12}$ Mm m$^2$ cm$^{-3}$) are derived from the extended AERONET data analysis described in Sects. 2 and 3.1 for all sites listed in Table 3. $c_{100,d}$ (in cm$^{-3}$ for $\sigma_d = 1$ Mm$^{-1}$), and $x_d$ and respective standard deviations (SD) are obtained in the way described in Sect. 3.2 by considering only AOT from 0.1-0.6, except for Ilorin (all AOT are used because only 12% of AOT<0.6). No data ($c_{100,d}$, $x_d$) are listed when the regression coefficient <0.6. The regional/continental mean values (for North Africa, Middle East, Asia, America/Australia) are obtained by observation-weighted averaging of the given station mean and SD values.

| Site | $c_{v,d}$ | $c_{v,df}$ | $c_{v,dc}$ | $c_{250,d}$ | $c_{s,d}$ | $c_{s,100,d}$ | $c_{100,d}$ | $x_d$ |
|---|---|---|---|---|---|---|---|---|
| N. Africa | $0.68 \pm 0.08$ | $0.23 \pm 0.06$ | $0.83 \pm 0.09$ | $0.18 \pm 0.03$ | $2.47 \pm 0.61$ | $1.59 \pm 0.10$ | $5.53 \pm 0.55$ | $0.84 \pm 0.02$ |
| TA | $0.67 \pm 0.07$ | $0.24 \pm 0.02$ | $0.81 \pm 0.08$ | $0.18 \pm 0.03$ | $2.52 \pm 0.60$ | $1.59 \pm 0.09$ | $5.80 \pm 0.42$ | $0.79 \pm 0.01$ |
| IZ | $0.59 \pm 0.05$ | $0.22 \pm 0.05$ | $0.72 \pm 0.06$ | $0.20 \pm 0.02$ | $2.39 \pm 0.52$ | $1.54 \pm 0.06$ | $6.85 \pm 1.07$ | $0.73 \pm 0.03$ |
| CV | $0.64 \pm 0.07$ | $0.22 \pm 0.06$ | $0.79 \pm 0.08$ | $0.20 \pm 0.03$ | $2.24 \pm 0.55$ | $1.58 \pm 0.10$ | $1.24 \pm 0.13$ | $1.04 \pm 0.02$ |
| DK | $0.69 \pm 0.08$ | $0.23 \pm 0.07$ | $0.84 \pm 0.09$ | $0.18 \pm 0.03$ | $2.54 \pm 0.62$ | $1.60 \pm 0.11$ | $7.42 \pm 0.81$ | $0.78 \pm 0.02$ |
| BA | $0.72 \pm 0.09$ | $0.24 \pm 0.07$ | $0.89 \pm 0.11$ | $0.18 \pm 0.03$ | $2.49 \pm 0.63$ | $1.60 \pm 0.11$ | $6.69 \pm 0.60$ | $0.80 \pm 0.02$ |
| IL | $0.73 \pm 0.11$ | $0.28 \pm 0.08$ | $0.91 \pm 0.15$ | $0.18 \pm 0.03$ | $3.04 \pm 0.90$ | $1.60 \pm 0.11$ | $4.52 \pm 1.15$ | $0.88 \pm 0.04$ |
| Middle East | $0.71 \pm 0.08$ | $0.24 \pm 0.07$ | $0.86 \pm 0.10$ | $0.16 \pm 0.02$ | $2.63 \pm 0.66$ | $1.58 \pm 0.10$ | $9.89 \pm 1.12$ | $0.73 \pm 0.02$ |
| EI | $0.67 \pm 0.09$ | $0.21 \pm 0.05$ | $0.83 \pm 0.10$ | $0.16 \pm 0.03$ | $2.40 \pm 0.51$ | $1.60 \pm 0.09$ | $10.74 \pm 2.11$ | $0.70 \pm 0.03$ |
| SB | $0.66 \pm 0.09$ | $0.23 \pm 0.06$ | $0.81 \pm 0.11$ | $0.18 \pm 0.03$ | $2.54 \pm 0.61$ | $1.57 \pm 0.09$ | $8.32 \pm 0.92$ | $0.73 \pm 0.02$ |
| NZ | $0.65 \pm 0.08$ | $0.24 \pm 0.08$ | $0.80 \pm 0.09$ | $0.18 \pm 0.03$ | $2.76 \pm 0.95$ | $1.56 \pm 0.10$ | $7.84 \pm 2.20$ | $0.75 \pm 0.05$ |
| SV | $0.74 \pm 0.08$ | $0.24 \pm 0.07$ | $0.90 \pm 0.10$ | $0.15 \pm 0.02$ | $2.66 \pm 0.69$ | $1.58 \pm 0.10$ | $11.98 \pm 1.06$ | $0.69 \pm 0.02$ |
| ME | $0.69 \pm 0.08$ | $0.24 \pm 0.06$ | $0.85 \pm 0.11$ | $0.16 \pm 0.02$ | $2.77 \pm 0.62$ | $1.60 \pm 0.09$ | $4.27 \pm 0.65$ | $0.89 \pm 0.03$ |
| Asia | $0.78 \pm 0.10$ | $0.27 \pm 0.08$ | $0.95 \pm 0.12$ | $0.14 \pm 0.03$ | $3.05 \pm 0.86$ | $1.57 \pm 0.09$ | $12.29 \pm 3.97$ | $0.71 \pm 0.05$ |
| DU | $0.79 \pm 0.09$ | $0.27 \pm 0.08$ | $0.96 \pm 0.12$ | $0.13 \pm 0.03$ | $3.11 \pm 0.87$ | $1.58 \pm 0.10$ | $12.36 \pm 3.49$ | $0.71 \pm 0.05$ |
| LA | $0.77 \pm 0.09$ | $0.27 \pm 0.09$ | $0.94 \pm 0.11$ | $0.15 \pm 0.02$ | $3.10 \pm 0.94$ | $1.54 \pm 0.08$ | $12.20 \pm 4.70$ | $0.70 \pm 0.06$ |
| DA | $0.73 \pm 0.17$ | $0.21 \pm 0.05$ | $0.92 \pm 0.19$ | $0.15 \pm 0.04$ | $2.45 \pm 0.48$ | $1.62 \pm 0.09$ | – | – |
| Amer./Aus. | $0.89 \pm 0.13$ | $0.23 \pm 0.06$ | $1.07 \pm 0.14$ | $0.11 \pm 0.03$ | $2.39 \pm 0.42$ | $1.64 \pm 0.11$ | $7.71 \pm 5.72$ | $0.73 \pm 0.13$ |
| TU | $0.79 \pm 0.15$ | $0.22 \pm 0.04$ | $0.98 \pm 0.17$ | $0.13 \pm 0.03$ | $2.36 \pm 0.22$ | $1.73 \pm 0.09$ | $4.57 \pm 5.10$ | $0.88 \pm 019$ |
| WS | $0.94 \pm 0.12$ | $0.22 \pm 0.05$ | $1.11 \pm 0.12$ | $0.10 \pm 0.03$ | $2.25 \pm 0.24$ | $1.60 \pm 0.06$ | – | – |
| TR | $0.89 \pm 0.12$ | $0.22 \pm 0.07$ | $1.08 \pm 0.14$ | $0.13 \pm 0.03$ | $2.47 \pm 0.73$ | $1.60 \pm 0.10$ | – | – |
| BI | $0.90 \pm 0.13$ | $0.25 \pm 0.07$ | $1.07 \pm 0.15$ | $0.11 \pm 0.03$ | $2.43 \pm 0.46$ | $1.62 \pm 0.12$ | $8.62 \pm 5.90$ | $0.69 \pm 0.12$ |
| GO | $0.62 \pm 0.08$ | $0.20 \pm 0.04$ | $0.81 \pm 0.09$ | $0.21 \pm 0.05$ | $2.25 \pm 0.46$ | $1.62 \pm 0.12$ | – | – |
| LI | $0.64 \pm 0.08$ | $0.27 \pm 0.08$ | $0.79 \pm 0.09$ | $0.18 \pm 0.03$ | $3.07 \pm 0.86$ | $1.63 \pm 0.19$ | – | – |

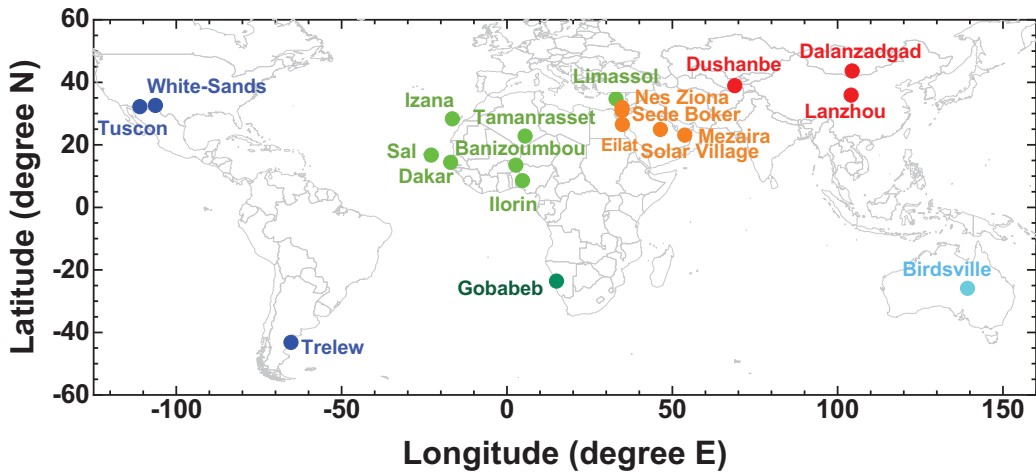

**Figure 1.** Overview of the 20 AERONET stations used in this study.

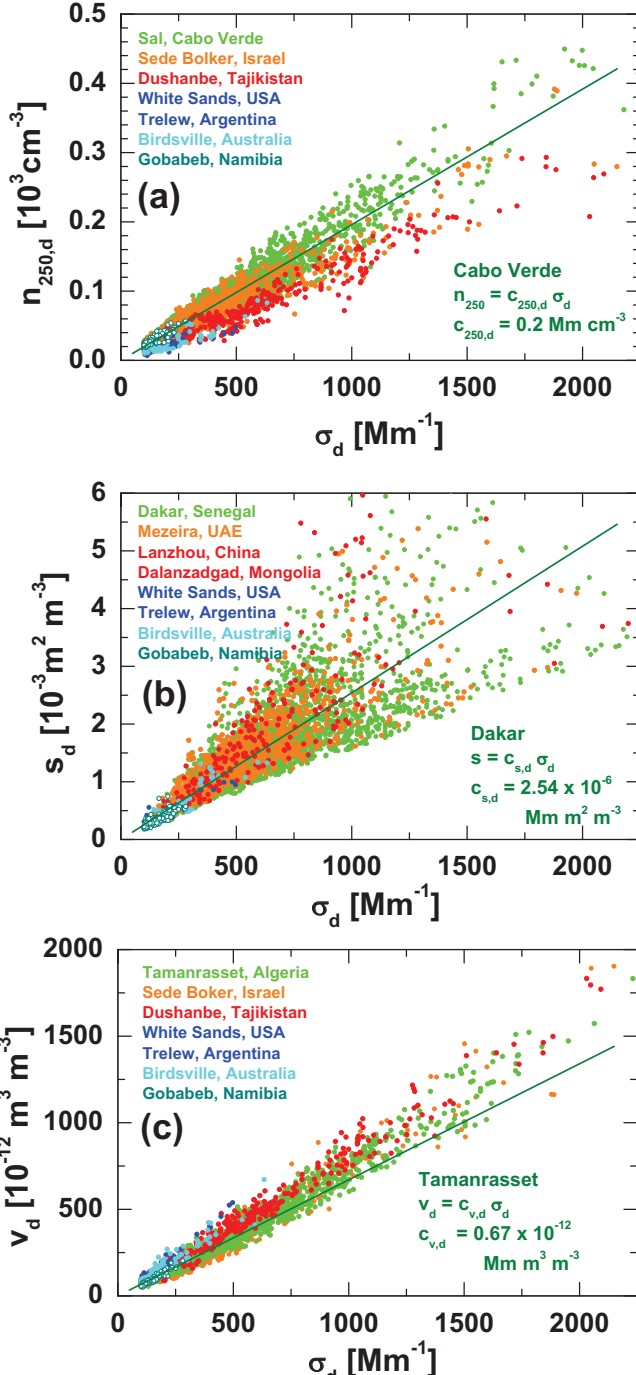

**Figure 2.** Relationship between dust extinction coefficient $\sigma_d$ (532 nm) and (a) dust particle number concentration $n_{250,d}$, (b) surface area concentration $s_d$, and (c) volume concentration $v_d$. Correlations are shown for dust-dominated AERONET data sets (AE< 0.3 and AOT>0.1 or $\sigma_d$ >100 Mm$^{-1}$) collected at sites in or close to major desert regions around the globe (indicated by different colors, see map in Fig. 1). The slopes of the dark green lines indicate the mean increase of $n_{250,d}$, $s_d$, and $v_d$ with $\sigma_d$ for the African stations as defined in Sect. 2 and thus the conversion factors $c_{250,d}$ (a, Eq. 9), $c_{s,d}$ (b, Eq. 10), and $c_{v,d}$ (c, Eq. 6), also given as numbers in (a), (b), and (c). All conversion parameters obtained from the entire AERONET analysis are listed in Table 4.

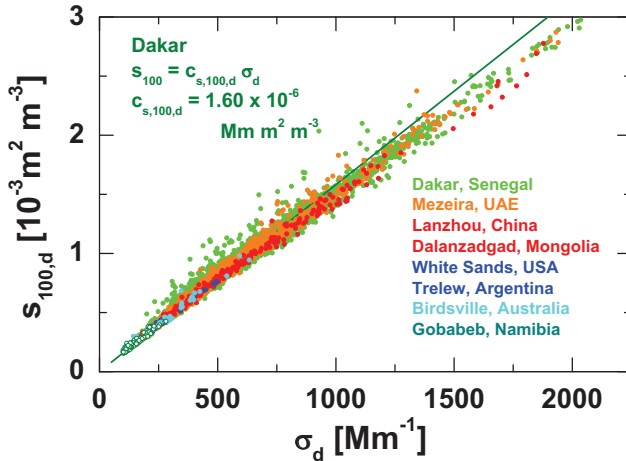

**Figure 3.** Same as Fig. 2, except for the relationship between dust extinction coefficient $\sigma_d$ (532 nm) and surface area concentration $s_{100,d}$ considering particles with radius >100 nm only. The slope of the dark green line indicates the mean increase of $s_{100,d}$ with $\sigma_d$ for Dakar, Senegal, as defined in Sect. 2 (Eq. 11).

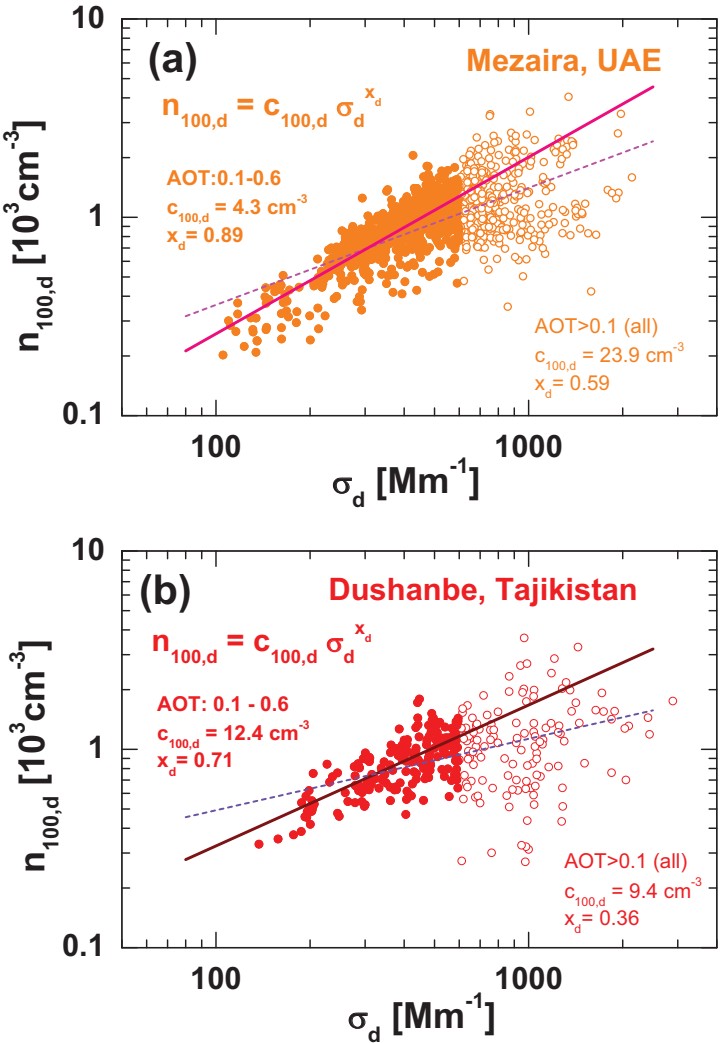

**Figure 4.** Relationship between dust extinction coefficient $\sigma_d$ (532 nm) and dust particle number concentration $n_{100,d}$ for AERONET dust observations at (a) Mezaira and (b) Dushanbe. Closed circles show the observations considering only 532 nm AOT values from 0.1-0.6. The open circles show all available observations (up to AOT of 3.0 or $\sigma_d$=3000 Mm$^{-1}$). The regression analysis is applied to the log($n_{100,d}$)-log($\sigma_d$) data field for each of the four data sets. The results of the analysis are given as numbers in the figures. The straight lines indicate the mean increase of log($n_{100,d}$) with log($\sigma_d$) and thus the $\sigma_d$ exponent $x_d$ (see Table 1, line 8, equation for $n_{100,d}$).

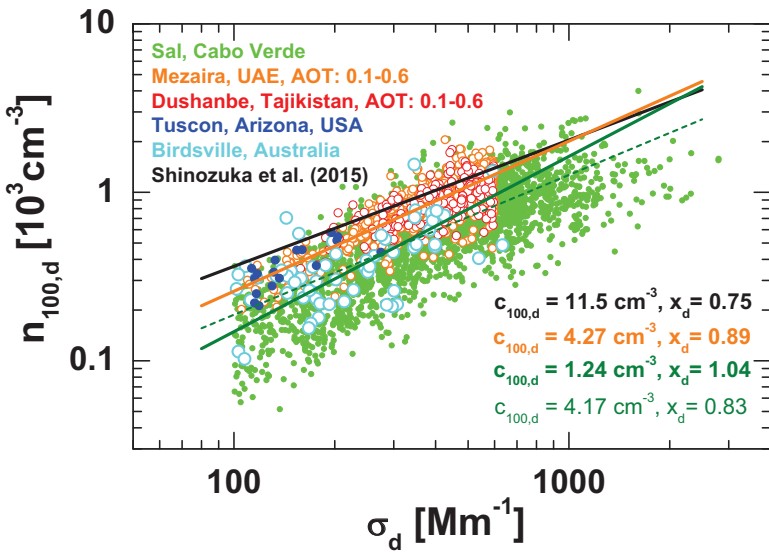

**Figure 5.** Relationship between dust extinction coefficient $\sigma_d$ (532 nm) and dust particle number concentration $n_{100,d}$ for different dust-dominated AERONET data sets collected at the indicated stations. The given regression analysis results and straight lines are based on observations at Mezaira (open orange circles, orange line, AOT from 0.1-0.6) and Cabo Verde (green closed circles, dark dashed green line, AOT from 0.1-3.0, and dark green thick solid line, AOT from 0.1-0.6). The dust conversion parameters presented by Shinozuka et al. (2015) are shown for comparison (black line and black $c_{100,d}$ and $x_d$ values).

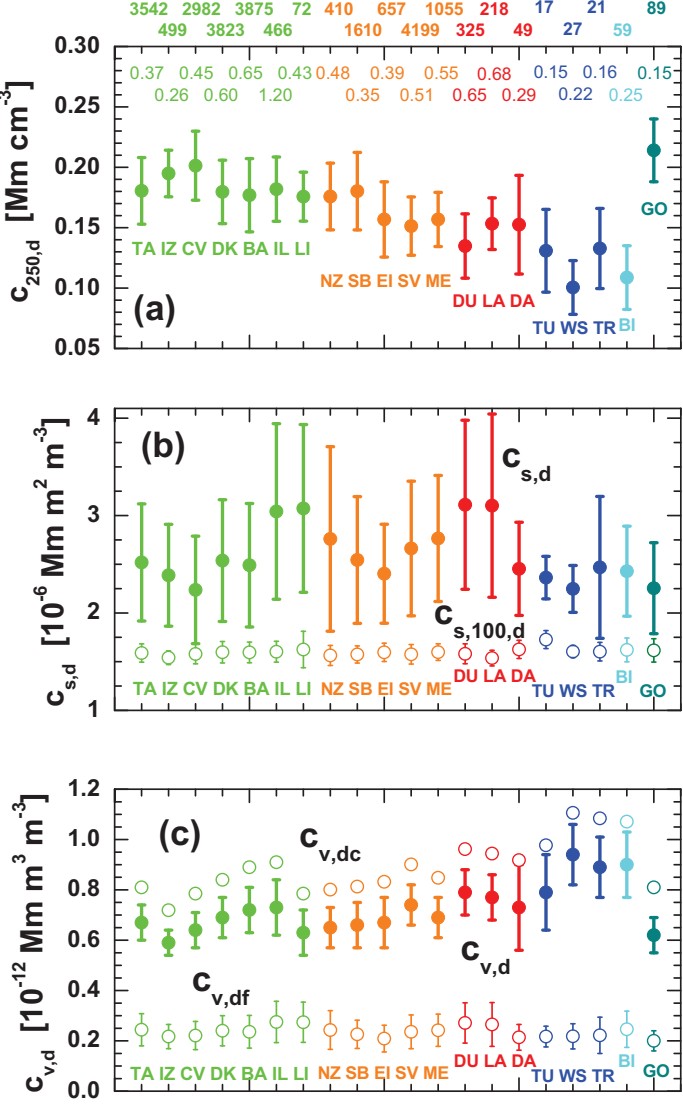

**Figure 6.** Overview of POLIPHON conversion factors (a) $c_{250,d}$, (b) $c_{s,d}$, and (c) $c_{v,d}$ (mean and SD) derived from AERONET dust data sets collected at 20 stations around the world. The stations (and acronmys) are given in Table 3. Total numbers of observations (considered in the statistical analyses for each stations) are given above the figure frame (a) followed by two lines with respective mean 532 nm dust AOTs for all data sets (considering only the dust cases with AOT>0.1). In (b), open circles indicate surface-related conversion factors considering particles with radius >100 nm, only. In (c), volume-related conversion factors are separately determined for total (index d), fine (index df, open symbols), and coarse dust (index dc, open symbols). The uncertainty bars for $c_{v,dc}$ are not shown, but similar to the ones for $c_{v,d}$. All statistical results are also summarized in Table 4.

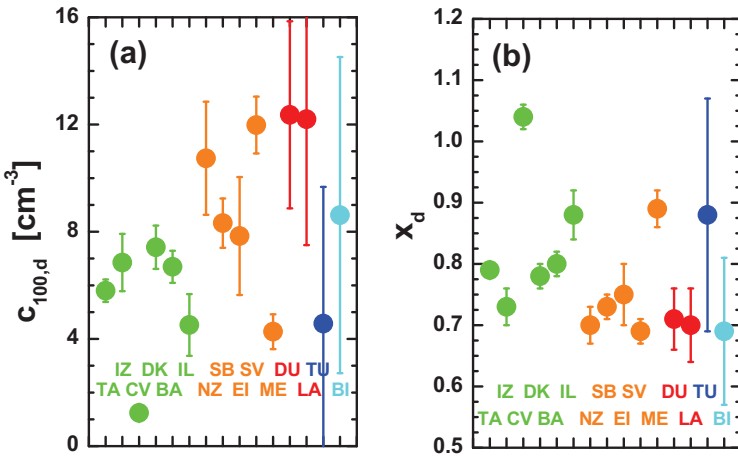

**Figure 7.** POLIPHON conversion parameters (a) $c_{100,d}$ and (b) $x_d$ derived from AERONET dust observations at 15 stations in northern Africa (green), the Middle East (orange), Central/East Asia (red), North America (blue), and Australia (light blue).

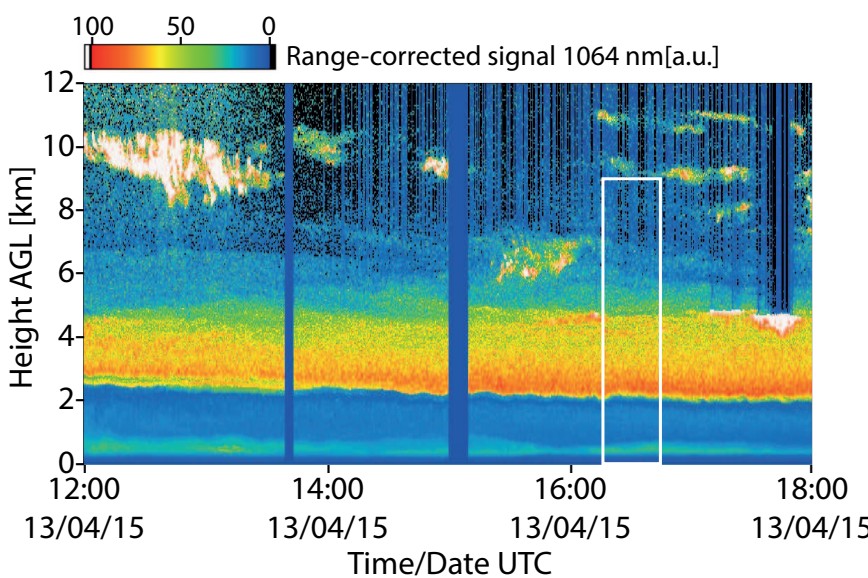

**Figure 8.** Dust layering over the Central Asian AERONET site of Dushanbe, Tajikistan, on 13 April 2015 observed with Polly lidar at 1064 nm (range-corrected signal). The densest layer from 2-5 km height AGL (above ground level) contained dust particles from Iran, Afganistan, and Oman according to the backward trajectories in Fig. 9. With increasing height, dust was advected from the Arabian peninsula and the Sahara. The polluted boundary layer reached up to about 2 km height and contained traces of local dust and dust from Kazakhstan. Above 6.5 km height (and temperatures $< -20°$C) ice clouds developed triggered by dust particles which are favorable ice-nucleating particles. POLIPHON results in Figs. 10-12 are derived for the height-time range indicated by the white rectangle.

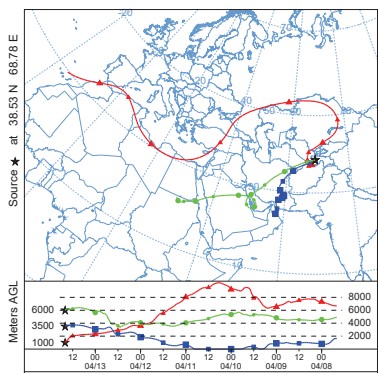

**Figure 9.** Six-day backward trajectories computed with the HYSPLIT (Hybrid Single Particle Lagrangian Integrated Trajectory) model (HYSPLIT, 2019; Stein et al., 2015; Rolph et al., 2017) for Dushanbe, Tajikistan, on 13 April 2015, 16:00 UTC. The computation is based on GDAS0.5 meteorological fields (GDAS, 2019). Arrival heights are at 1000 m (red, in the boundary layer with Central Asian dust), 3500 m (blue, in a dense layer with dust from several western Asian deserts), and 6000 m (green, in dusty air from the Arabian peninsula and the Sahara).

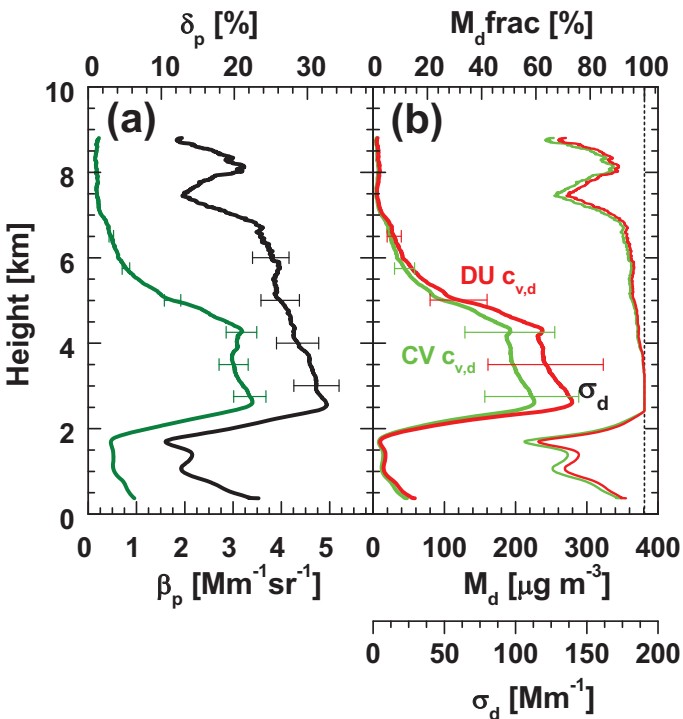

**Figure 10.** Retrieval of dust mass concentrations. From profiles of the particle backscatter coefficient $\beta_p$ (green curve in (a), 532 nm) and particle linear depolarization ratio $\delta_p$ (black curve in (a), 532 nm) the profile of the dust backscatter coefficient $\beta_d$ is determined and then converted into the dust extinction coefficient $\sigma_d$ (red curve in b) by means of a lidar ratio of 40 sr. The $\sigma_d$ profile is then converted into mass concentrations $M_d$ (shown in (b) as thick lines) by means of volume conversion factors $c_{v,d}$ of $0.64 \times 10^{-12}$ Mm for Sal, Cabo Verde (CV, green $M_d$ profile, see Table. 4) and $0.79 \times 10^{-12}$ Mm for Dushanbe (DU, red $M_d$ profile). Respective profiles of $M_d$ fraction (thin red and green curves in b) are also shown. The Polly lidar observation was performed at Dushanbe on 13 April 2015, 16:15-16:44 UTC (white rectangle in Fig. 8). The temporally averaged lidar signal profiles were smoothed with 750 m before the computation of $\beta_p$ and $\delta_p$. Error bars indicate (a) 10% and (b) 30% uncertainty (typical uncertainty according to Table 1).

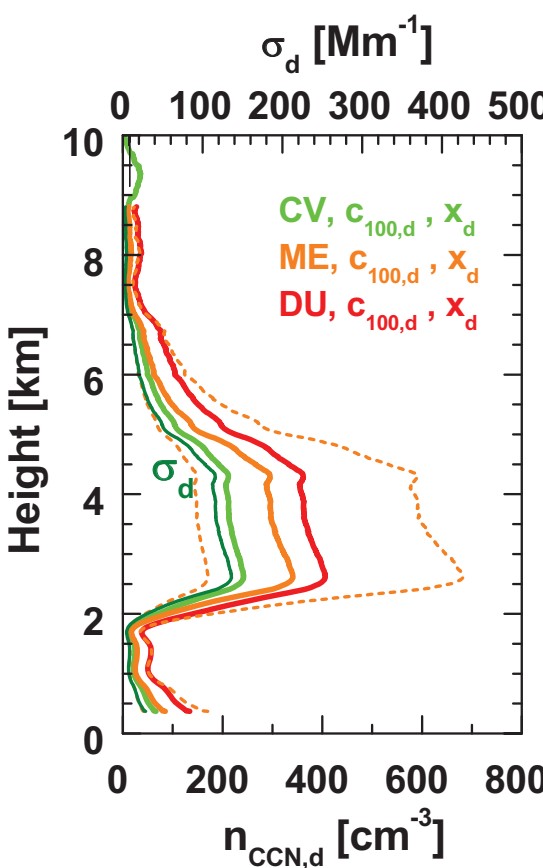

**Figure 11.** Estimation of dust CCNC profiles for the 16:15-16:44 UTC time period on 13 April 2015. The profile of the dust extinction coefficients $\sigma_{\mathrm{d}}$ (thin dark green) is converted into a profile of $n_{100,\mathrm{d}}$ by means the conversion parameters $c_{100,\mathrm{d}}$ and $x_{\mathrm{d}}$ given in Table 4 for Sal, Cabo Verde (CV, green profile), Mezaira (ME, orange profile), and Dushanbe (DU, red profile). For a typical water supersaturation value of 0.2% at the base of a convective cloud, $f_{ss,\mathrm{d}} = 1.0$ and thus $n_{100,\mathrm{d}} \approx n_{\mathrm{CCN},\mathrm{d}}$ (see Table 1, line 12, equation for $n_{\mathrm{CCN},ss,\mathrm{d}}$). The uncertainty range is assumed to be of the order of a factor of 2 (indicated by dashed curves around the orange ME curve).

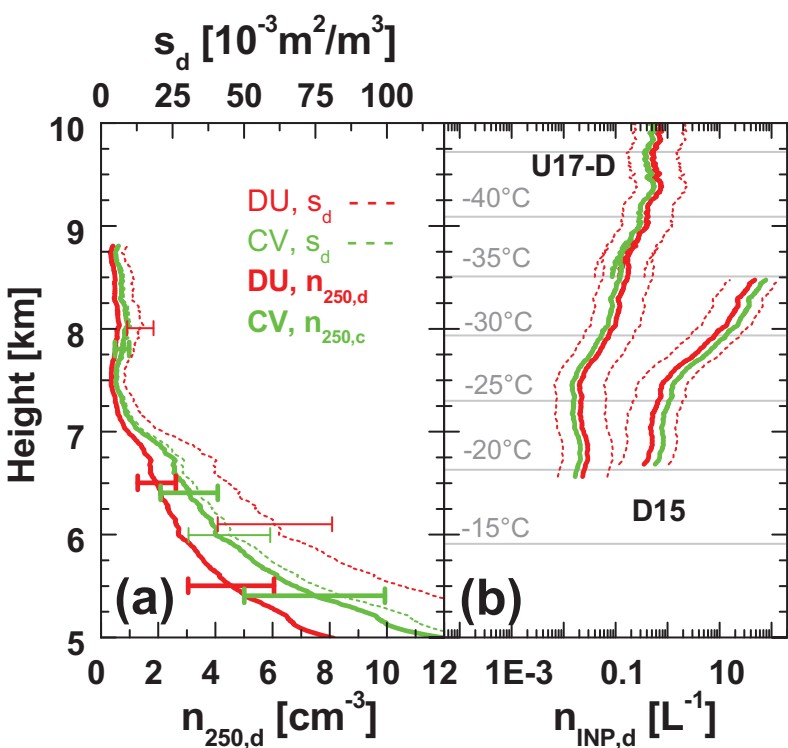

**Figure 12.** Estimation of INPC profiles for the 16:15-16:44 UTC time period on 13 April 2015. The profile of $\sigma_d$ in Fig. 11 (thin dark green profile) is converted into profiles of the particle number concentration $n_{250,d}$ (thick solid lines in a) and surface area concentration $s_d$ (thin dashed lines in a) by means of the conversion factors $c_{250,d}$ and $c_{s,d}$ in Table 4 for Dushanbe (DU, red profiles) and Sal, Cabo Verde (CV, green profiles). The profiles of $n_{250,d}$ and $s_d$ together with the actual GDAS temperature profile are input parameters in the INP parameterization schemes U17-D (deposition nucleation) and D15 (immersion freezing, see Table 1). In the deposition-nucleation INPC estimation, a typical ice supersaturation values of $S_{ice} = 1.15$ is assumed. The INP parameterizations are valid for temperatures of about $-20°C$ and lower. The D15 parameterization holds for temperatures down to $-35°C$ only. Error bars indicate uncertainties of 30% in (a). The $n_{INP,d}$ uncertainty range is assumed to be one order of magnitude indicated by dashed lines in (b). We added $n_{INP,d}$ profile segments for the 8.5-10 km height range, derived from lidar observations in cloud-free air from 15:15-16:10 UTC on 13 April 2015 (see Fig. 8).