# Peer review of "Dust mass, CCN, and INP profiling with polarization lidar: Updated POLIPHON conversion factors from global AERONET analysis"

_Atmospheric Measurement Techniques, 2019_

## Referee Comment (RC1) · Anonymous Referee #1 · 14 Apr 2019

The paper by Ansmann et al presents updated conversion factors for dust in the POLIPHON retrieval. The conversion factors are based on AERONET inversion results for a range of stations around the globe affected by desert dust. Overall the paper is well-written and useful and could be published in AMT after revisions.

General comments:

A validation with independent data would be very useful in general. I understand that such independent data sets that could be used for that purpose are not easy to obtain. Are such comparisons for same stations around the globe planned in the future?

The surface area concentration $s_d$ is used for an INPC estimation approach (page 3

[Figure]

/ line 17). It seems to me that it would be better not to use the complete AERONET size range starting at 50nm to calculate the surface area concentration, because of the following reasons:

1) Such small particles are probably not relevant for INPC (the other appoach considers only particles with r>=250nm probably because of this).

2) The AERONET observations are not really sensitive to aerosol in the first size bins.

3) A large fraction of these small particles (if they are no inversion artefact) are probably not mineral dust particles as shown in several studies, e.g. by Konrad Kandler.

To illustrate the importance of the small size bins for the surface area calculations, I created a plot (see attached Figure 1) with the surface area concentration calculated from the AERONET dust data measured in Dakar (dust cases, i.e. Angstrom<0.3 and $\tau_{532}$>0.1). The upper left plot (starting at first bin) corresponds to the Dakar data in Figure 1b of the discussion paper. If the first bins are not considered for the calculation of surface area the spread gets much smaller. When the surface area is calculated starting at bin 5 or 7 ($\approx$250nm) the data is almost on a line (middle right and lower left plot). The lower right plot of the attached figure shows the complete surface area versus the surface area starting at bin 7. It can be seen that sometimes the surface area for r>250nm (where dust is usually dominating in desert aerosols) is only one fifth of the complete surface area in the AERONET data set. Though a factor of five is still less than an order of magnitude (as taken into account by the authors in the discussion) I think it is worth to take into consideration the minimum radius in the surface area calculations. In my view, r>=250nm would make much more sense than 50nm for the reasons given above. Maybe the authors want to discuss this.

Minor corrections:

Page 1 / line 4: "miccrophysical" –> "microphysical"

Page 1 / line 22: "separation of dust from aerosol pollution optical properties" is a bit confusing. Please rephrase.

Page 2 / line 14: "The technique is based on the conversion of lidar-derived particle extinction coefficients into ...": As far as I understand POLIPHON uses backscatter coefficients (+ depol) as input (also shown in Fig. 8). Therefore I think "extinction coefficient" should be exchanged here by "backscatter coefficient".

Page 5 / line 6: After "21 AERONET station" a reference to Tab. 2 should be added. Otherwise one asks at this point: Which 21 stations?

Page 5 / line 20: "enough" should be removed.

Page 7 / line 12: "inside" –> "insight"

Page 8 / line 14: I think the unit here should be $cm^{-3}$ not $cm^{-1}$.

Page 8 / line 28-30 and Fig 7b: The part about the forward trajectories is in principle interesting but I am not sure if it fits very well here as it may confuse the reader and leaves some questions. For example, is there some wash-out during the further transport?

Page 10 / line 8: "sets" –> "set"

Table 1: In the line with $n_{100,d}(z)$: Shouldn't $\sigma_d$ not be divided by some "normalization extinction coefficient", for example that "$\sigma_d^{x_d}(z)$" gets "$(\frac{\sigma_d}{1Mm^{-1}})^{x_d}(z)$"? Otherwise the units don't make sense.

Figure 10: "$c_{250,d}$" and "$c_{s,d}$" in the figure probably could be removed.
* * *
[Figure]

**Fig. 1.** Surface area concentrations from AERONET inversions for 'dust cases' at Dakar.

---

## Referee Comment (RC2) · Anonymous Referee #2 · 23 Apr 2019

The paper "Dust mass, CCN, and INP profiling with polarization lidar: Updated POLIPHON conversion factors from global AERONET analysis" presents and discusses the POLIPHON methodology, implemented with an updated set of dust conversion factors, considering the different dust sources around the globe. These different conversion parameters are of critical importance for the POLIPHON methodology in order to compute dust mass, CCN and INP profiles. The paper is not only limited to computing, providing and discussing the novel set of dust conversion factors. A complex case of dust advection over the EARLINET-Dushanbe station in Tajikistan, originating from different desert dust sources is presented. The study falls within the scope of AMT. The authors have done a thorough job, the manuscript is well-written

[Figure]

/ structured, the presentation clear, the language fluent and the quality of the figures high. Furthermore, the authors give credit to related work and the results support the conclusions. However, in order to help improving the manuscript, I would kindly suggest the authors to take into account the following specific comments.

1. The authors refer to the use of the "AERONET data base" in the manuscript. I suggest to provide more detailed information regarding AERONET (e.g. Version) and the use of AERONET data (e.g. level, files, name/list of parameters, units) in POLIPHON method. This is only done in Table 2 (version 3, level 2.0) but I believe it would be useful for the reader if it also stated in the manuscript.

2. I would recommend the authors to use a "world map" figure in the introduction, to give the reader an overview of the AERONET stations used in the study.

3. The manuscript provides a novel dataset of conversion factors for desert dust originating from different dust sources. Table 1 provides the input parameters in the POLIPHON method. However, no discussion is provided regarding the dust extinction coefficient. For instance, desert dust sources around the globe are characterized by different extinction-to-backscatter ratio and in addition different dust particulate depolarization ratio. Since the manuscript aims to provide the POLIPHON conversion factors per different station, has the different dust source per observed case been considered? For instance, how are the observed cases in Dushanbe, where dust originates from different sources as shown, were treated in terms of input dust extinction coefficient values? I assume that the authors have used proper inputs of lidar ratio and dust depolarization per different desert. Thus I would recommend the authors to provide a thorough discussion on the used parameters and in addition a table of the different values used in the methodology, since the accurate computation of the dust extinction coefficient is a critical input for the POLIPHON method per desert region. Have the authors considered the use of HYSPLIT in order to quantify the effect of different desert sources in the computation of the conversion factors for each AERONET station, in order to attribute the provided conversion values in the present manuscript

not confined locally, to a station, but to extend the conversion factors to larger regions?

4. Page 5 – Conversion parameters from the AERONET data base. The authors state at the same time that "We preferred stations with long data records and large numbers of observations ... for the statistical analysis" and that "We added the Leipzig AERONET observations with a small number of strong Saharan dust outbreaks"– but also stations like "Tuscon, Arizona" (17 dust observations), "White-Sands" (27 dust observations) and "Trelew, Argentina" (21 dust observations). Please consider revising the paragraph, since although the first statement holds for most of the AERONET stations it contradicts the use of other stations in the manuscript.

5. The authors are limiting the available AERONET measurements to dust dominated cases by defining all useful cases to have an Angstrom Exponent (AE) value less than 0.3 and Aerosol Optical Thickness (AOT) value larger than 0.1. My consideration mainly applies to near-coastal regions. Have the authors somehow tried to exclude the marine particles contribution to these cases? Are additional parameters considered when the dust dominated cases are selected? (i.e. the spectral dependence of the SSA?)

6. Table 1. Please consider expanding the Table to include units for the input and the output parameters, while at the same time the use of an additional column with the computed uncertainties (used also in the manuscript) per output parameter would be helpful for a potential user of the POLIPHON method.

7. Table 2. Please consider expanding the Table to include not only Dust AOT but the Total AOT, since the authors provide in addition to dust observations the total number of AERONET observations (dust and non-dust).

8. The authors provide the POLIPHON conversion factors in figures 4 and 5. I suggest the authors to include (on parallel to these figures and per conversion factor), world maps of the AERONET sites used in the study with the computed conversion factors with different color, depending on the computed conversion values, to demonstrate

more clear the spatial distribution of the provided values.

9. Figure 8 and Figure 11. The authors use error-bars in the figures as a metric of the uncertainty, however it is not clear in the manuscript whether the shown uncertainties are computed for the shown cases, or are the more generic uncertainties computed and discussed in previous POLIPHON papers.

10. I suggest the authors to delineate the desert domains related to each AERONET site provided in Table 3, in order to facilitate the use of the conversion factors provided in the manuscript for global studies.

11. Page 2, Line 12: "ice and precipitation formation already at high temperatures of -15 to -35 C". Please provide relevant references.

12. Page 3, Line 6: It is not clear to the reader what the parameter fss stands for. Same also holds for Table 1. Please revise accordingly.

13. Please provide more information on the temperature values used as input for the INP retrievals with the D15 and U17 schemes. Are those data provided from local radiosondes?

14. Page 4, equations 7 and 8: there is a typo, the values udf, j and udc, j should be in reverse in the two equations.

15. Page 4, Line 11: "with the conversion factor $c_{v,i,\lambda}$ and the particle extinction co-efficient $\sigma_{i,\lambda}$ measured with lidar at wavelength $\lambda$" Please rephrase so that it is more clear, even to the less experienced reader that the conversion factor is not provided by lidar but from the AERONET measurements.

16. Page 6, Line 4: "by dividing", do the authors mean by multiplying?
* * *

---

## Referee Comment (RC3) · Anonymous Referee #3 · 3 May 2019

Review of the paper by Ansmann et al.,

The paper presents an update of the POLIPHON method, which has been already introduced by the authors in a series of papers. In the current paper the authors provide a very comprehensive overview of the method. The updates presented concern the potential applicability of the method on a global scale and thus the application of POLIPHON on satellite lidar data. The paper is well written and structured and should be accepted for publication in AMT after considering few remarks listed below.

Comments:

Page 2, line 22. The authors should also mention that their study is relevant not only to

[Figure]

PollyNET but also to lidar networks with long-term measurements and well established QA procedures (e.g. EARLINET)

Page 5, line 27. All stations selected from AERONET correspond to stations in the proximity of deserts, except Leipzig. The inclusion of Leipzig can confuse the reader. What is the significance for the inclusion of Leipzig. If the authors would be interested to examine the possible variability of the conversion factors as function of the distance from the source, then they should examine also other AERONET stations, with variable distances from the desert (there are plenty in the Mediterranean). Please comment.

Page 7, lines 12-17. Is it possible that in Capo Verde one might still expect the influence of smoke particles in large AOTs?

Page 7, line 21. The authors should make a comment here why they think that a product with an overall error of a factor 2-3 is useful and relevant.

Page 8, line 19. The authors probably mean "selected" rather than "elected".

Page 8, line 22. How do the authors distinguish at 10km dust from cirrus?

Page 8, line 25-30. The trajectory analysis provides some indication for the origin of the observed layers. Are there any model simulations available that confirm and further support this multi-source structure? The inclusion and discussion of figure 7b, to my view could be omitted. It just opens a new discussion, which is left incomplete.

Conclusions. (page 10, lines 14-17). This statement is confusing as written. The authors first they suggest to use globally valid conversion factors and then recommend to use regional ones. Maybe each suggestion should be followed with an uncertainty estimate. Please consider to rephrase the recommendations, since these are the ones to be followed by a potential user of POLIPHON.

---

## Author Comment (AC1) · 5 Jul 2019

Dear Editor, dear reviewers!

The following letter includes our reply to all comments of the three reviewers.

We thank the reviewers for carefully reading and for making good suggestions. We considered almost all of them. Our answers are in blue.

Before we provide an item by item reply, in the beginning, a list of main changes and improvements:

- All tables are improved considering the suggestion of the reviewers.
- New Table 2 with lidar ratios for different dust source regions is added.
- Discussion on dust depolarization ratio and lidar ratios is added.
- New Fig. 1 (global map with 20 AERONET stations) is added,
- New Fig. 3, showing the relationship between surface area s100d (for particles with radius>100nm) and extinction coefficient, is added.
- s100d retrieval is included in the methodology, in Table 4, and in the discussion.
- The presentation of the methodology in Sect. 2 is simplified. Only focus on dust (no aerosol-type-dependent retrieval with aerosol-type index i anymore).

All significant changes will be highlighted in **BOLD** in the revised version that is almost finalized.

**Reviewer #1**

General comments:

A validation with independent data would be very useful in general. I understand that such independent data sets that could be used for that purpose are not easy to obtain. Are such comparisons for same stations around the globe planned in the future?

The AERONET data set is unique. There is no alternative! One needs consistent data sets that cover both optical as well as microphysical properties. And such a consistency will never be available in case of independent in situ measurements of optical and microphysical properties. Both will have their specific not well known and characterized biases. As a constructive alternative, we check the accuracy of the POLIPHON products by comparing the products with independent observations of dust mass concentrations, of CCN concentration, and the parameters need to estimate INP concentration. These efforts (plus references are given in the introduction.

The surface area concentration $s_d$ is used for an INPC estimation approach (page 3/ line 17). It seems to me that it would be better not to use the complete AERONET size range starting at 50nm to calculate the surface area concentration, because of the following reasons:

1) Such small particles are probably not relevant for INPC (the other appoach considers only particles with r>=250nm probably because of this).

2) The AERONET observations are not really sensitive to aerosol in the first size bins.

3) A large fraction of these small particles (if they are no inversion artefact) are probably not mineral dust particles as shown in several studies, …

… I think it is worth to take into consideration the minimum radius in the surface area calculations. In my view, r>=250nm would make much more sense than 50nm for the reasons given above. Maybe the authors want to discuss this.

We thank the reviewer for the nice study! Good idea! Nevertheless: In the laboratory studies (Ullrich et al., 2017) they count the nucleated ice crystals in the AIDA chamber (KIT, Karlsruhe) and measured at the same time the dust size distribution and in this way the overall surface area concentration. So, the INP parameterization  is linked to the total surface area concentration. We cannot deviate from this approach! However, and this is now added (Figure 3 in the revised version), we make an attempt to use the surface area which considers only the CCN dust particles, that means, particles with radius larger than 100nm. This is justified in case of immersion freezing where the CCN used to form droplets are later on the INP, and provide the surface-area for ice nucleation. This aspect is now included in the paper. On the other hand, all the dust field observations show that there are always dust particles even with sizes as low as 50 nm. We cannot neglect that. But, if we use an Angstroem exponent of 0.3 as the upper limit, we are sure that the anthropogenic pollution impact is negligible. We extended the discussion on this in the paper a bit.

Minor corrections:

Page 1 / line 4: "miccrophysical" –> "microphysical"

Done

Page 1 / line 22: "separation of dust from aerosol pollution optical properties" is a bit confusing. Please rephrase.

Done

Page 2 / line 14: "The technique is based on the conversion of lidar-derived particle extinction coefficients into ...": As far as I understand POLIPHON uses backscatter coefficients (+ depol) as input (also shown in Fig. 8). Therefore I think "extinction coefficient" should be exchanged here by "backscatter coefficient".

Done, rephrased

Page 5 / line 6: After "21 AERONET station" a reference to Tab. 2 should be added. Otherwise one asks at this point: Which 21 stations?

Done (now we have 20 stations, the Leipzig station and data are removed from the paper)

Page 5 / line 20: "enough" should be removed.

Done

Page 7 / line 12: "inside" –> "insight"

Done

Page 8 / line 14: I think the unit here should be $cm^{-3}$ not $cm^{-1}$.

Done

Page 8 / line 28-30 and Fig 7b: The part about the forward trajectories is in principle interesting but I am not sure if it fits very well here as it may confuse the reader and leaves some questions. For example, is there some washout during the further transport?

Removed

Page 10 / line 8: "sets" –> "set"

Done

Table 1: In the line with $n_{100,d}(z)$: Shouldn't $\sigma_d$ not be divided by some "normalization extinction coefficient", for example that "$\sigma x_d(z)$" gets "$(\sigma_{d}1Mm^{-1})x_d(z)$"? Otherwise the units don't make sense.

Done

Figure 10: "$c_{250,d}$" and "$c_{s,d}$" in the figure probably could be removed.

Changed

**Reviewer #2**

… I would kindly suggest the authors to take into account the following specific comments.

1. The authors refer to the use of the "AERONET data base" in the manuscript. I suggest to provide more detailed information regarding AERONET (e.g. Version) and the use of AERONET data (e.g. level, files, name/list of parameters, units) in POLIPHON method. This is only done in Table 2 (version 3, level 2.0) but I believe it would be useful for the reader if it also stated in the manuscript.

Improved (in Sect.2 and Sect.3)

2. I would recommend the authors to use a "world map" figure in the introduction, to give the reader an overview of the AERONET stations used in the study.

Done (Figure 1 in the revised version)

3. The manuscript provides a novel dataset of conversion factors for desert dust originating from different dust sources. Table 1 provides the input parameters in the POLIPHON method. However, no discussion is provided regarding the dust extinction coefficient. For instance, desert dust sources around the globe are characterized by different extinction-to-backscatter ratio and in addition different dust particulate de-polarization ratio. Since the manuscript aims to provide the POLIPHON conversion factors per different station, has the different dust source per observed case been considered? For instance, how are the observed cases in Dushanbe, where dust originates from different sources as shown, were treated in terms of input dust extinction coefficient values? I assume that the authors have used proper inputs of lidar ratio and dust depolarization per different desert. Thus I would recommend the authors to provide a thorough discussion on the used parameters and in addition a table of the different values used in the methodology, since the accurate computation of the dust extinction coefficient is a critical input for the POLIPHON method per desert region. Have the authors considered the use of HYSPLIT in order to quantify the effect of different desert sources in the computation of the conversion factors for each AERONET station, in order to attribute the provided conversion values in the present manuscript not confined locally, to a station, but to extend the conversion factors to larger regions?

We extended the discussion on depolarization ratios and lidar ratios for different dust source regions (new Table 2 with lidar ratios and references), and we use different conversion factors in the Dushanbe case study to show the impact of uncertainties in the retrieval products (as shown in the figures in the Dushanbe section). But we leave out to use trajectory-based selected different conversion parameters. If we would have to use trajectories to decide what conversion factors we should use (to avoid large uncertainties) then the method is no longer attractive. The method would be too complicated. We want to keep everything as simple as possible. We feel it is not necessary to switch always the conversion parameters. Better to use just two very different conversion parameter sets as shown in the case study section, Sect. 4, to indicate the uncertainty range. The tables with all the numbers for different deserts are available in the paper for that. Of course, we check trajectories to understand the observations and the long-range transport of dust. And besides HYSPLIT we frequently also used more sophisticated transport models (FLEXPART), also to get an idea about the HYSPLIT trajectory uncertainties.

4. Page 5 – Conversion parameters from the AERONET data base. The authors state at the same time that "We preferred stations with long data records and large numbers of observations...for the statistical analysis" and that "We added the Leipzig AERONET observations with a small number of strong Saharan dust outbreaks"– butalso stations like "Tuscon, Arizona" (17 dust observations), "White-Sands" (27 dust observations) and "Trelew, Argentina" (21 dust observations). Please consider revising the paragraph, since although the first statement holds for most of the AERONET stations it contradicts the use of other stations in the manuscript.

This is now rephrased, and the Leipzig (site, data, conversion factors) are totally removed from the paper.

5. The authors are limiting the available AERONET measurements to dust dominated cases by defining all useful cases to have an Angstrom Exponent (AE) value less than 0.3 and Aerosol Optical Thickness (AOT) value larger than 0.1. My consideration mainly applies to near-coastal regions. Have the authors somehow tried to exclude the marine particles contribution to these cases? Are additional parameters considered when the dust dominated cases are selected? (i.e. the spectral dependence of the SSA?)

No! We only considered AOT>0.1 and AE<0.3. Again we want to keep the criteria as simple as possible, robust, and rather basic (no sophisticated retrieval product..). We discuss now the potential impact of marine particles. AOT>0.1 is already introduced to remove the marine cases and to reduce the remaining marine effects (typical marine AOT is 0.05). Furthermore, the dust AOT statistics (mean, SD) in Table 3 indicate that the marine AOT impact is generally low. And at the end the size distribution of dust and sea salt is quite similar….and thus the conversion parameters are not so different (we avoid to discuss this point in the paper to make it not too complicated). So the overall effect of marine particles is low.

6. Table 1. Please consider expanding the Table to include units for the input and the output parameters, while at the same time the use of an additional column with the computed uncertainties (used also in the manuscript) per output parameter would be helpful for a potential user of the POLIPHON method.

We include dimensions for the retrieval products and state in the figure caption the dimensions for the backscatter and the extinction coefficient (the main lidar input). There is also a new column with uncertainty ranges from typical to extreme uncertainties.

7. Table 2. Please consider expanding the Table to include not only Dust AOT but the Total AOT, since the authors provide in addition to dust observations the total number of AERONET observations (dust and non-dust).

Done

8. The authors provide the POLIPHON conversion factors in figures 4 and 5. I suggest the authors to include (on parallel to these figures and per conversion factor), world maps of the AERONET sites used in the study with the computed conversion factors with different color, depending on the computed conversion values, to demonstrate more clear the spatial distribution of the provided values.

We prefer the compact figures to see differences and therefore di not follow the suggestion of the reviewer. We think that the new Fig. 1 (the world map with AERONET stations used) is sufficient. The different dust source regions are indicated by different colors and that helps already a lot. And if we would put numbers in a world map, a comparison would no longer be so easy as in Fig. 4 (new Fig. 6). Figure 5 (new Fig. 7) is just good to see how variable the conversion parameters are in case of the n100d retrieval.

9. Figure 8 and Figure 11. The authors use error-bars in the figures as a metric of the uncertainty, however it is not clear in the manuscript whether the shown uncertainties are computed for the shown cases, or are the more generic uncertainties computed and discussed in previous POLIPHON papers.

More generic! Rough estimates.

10. I suggest the authors to delineate the desert domains related to each AERONETsite provided in Table 3, in order to facilitate the use of the conversion factors provided in the manuscript for global studies.

We do not like the idea! Furthermore, we omitted the global conversion parameter sets (from Table 4). We leave it open to the reader how to use the conversion factors. We suggest to use just two different conversion parameter sets to produce something like a solution space (min and max profiles) and in this way to characterize the uncertainty introduced by the conversion procedure. We state that in Sect.4.

11. Page 2, Line 12: "ice and precipitation formation already at high temperatures of-15 to -35 C". Please provide relevant references.

Done, Seifert et al., 2010

12. Page 3, Line 6: It is not clear to the reader what the parameter fss stands for. Same also holds for Table 1. Please revise accordingly.

Done

13. Please provide more information on the temperature values used as input for theINP retrievals with the D15 and U17 schemes. Are those data provided from local radiosondes?

Done. GDAS data are used instead of radiosonde profiles. GDAS data consider all radiosonde ascents, worldwide. The GDAS data are much better than individual radiosonde profiles.

14. Page 4, equations 7 and 8: there is a typo, the values udf, j and udc, j should be in reverse in the two equations.

Improved

15. Page 4, Line 11: "with the conversion factor cv,i,λ and the particle extinction co-efficient σi,λ measured with lidar at wavelength λ" Please rephrase so that it is more clear, even to the less experienced reader that the conversion factor is not provided by lidar but from the AERONET measurements.

Done

16. Page 6, Line 4: "by dividing", do the authors mean by multiplying?

No! Divided is correct. But to avoid confusion, we rephrased the sentence.

**Reviewer #3**

Comments:

Page 2, line 22. The authors should also mention that their study is relevant not only to PollyNET but also to lidar networks with long-term measurements and well established QA procedures (e.g. EARLINET).

Done

Page 5, line 27. All stations selected from AERONET correspond to stations in the proximity of deserts, except Leipzig. The inclusion of Leipzig can confuse the reader. What is the significance for the inclusion of Leipzig. If the authors would be interested to examine the possible variability of the conversion factors as function of the distance from the source, then they should examine also other AERONET stations, with variable distances from the desert (there are plenty in the Mediterranean). Please comment.

Leipzig data are completely removed from the paper. 20 AERONET stations are left.

Page 7, lines 12-17. Is it possible that in Capo Verde one might still expect the influence of smoke particles in large AOTs?

No! Our experience with Angstroem exponents is that such low values of 0.3 as we use as upper limited of considered dust cases does not leave room for any significant fine mode pollution impact. If pollution is present and sensitive to influence the optical properties, the Angstroem exponent would increase immediately to 0.6 and more. We removed sentences (given in the submitted version of the manuscript) that fine mode pollution may have influenced the

conversion factor determination. This is a misleading statement. For Angstroem exponents <0.3, this is impossible.

Page 7, line 21. The authors should make a comment here why they think that a product with an overall error of a factor 2-3 is useful and relevant.

We extended the discussion and state something like this: Meanwhile we have several CCNC and INPC comparison studies. These studies indicate uncertainties of the order of 50%. On the other hand we also state that uncertainties of a factor of 2-3 are acceptable in long-term climatological studies. It is even better to have uncertain observations and derived statistics than having nothing.

Page 8, line 19. The authors probably mean "selected" rather than "elected".

Yes

Page 8, line 22. How do the authors distinguish at 10km dust from cirrus?

Dust depol ratio is always <35%, cirrus depol ratio is always >40%. And cirrus (or more general clouds) causes sharp changes in backscatter (in height and time) that is not the case for aerosols (always comparably smooth structures). So, the combination of backscatter and depolarization helps to distinguish. We state that in Sect. 4.

Page 8, line 25-30. The trajectory analysis provides some indication for the origin of the observed layers. Are there any model simulations available that confirm and further support this multi-source structure? The inclusion and discussion of figure 7b, to my view could be omitted. It just opens a new discussion, which is left incomplete.

Agree! We removed the old Fig. 7b (forward trajectories). And we use FLEXPART in some case studies to get more insight in the atmospheric transport conditions, but also to check the HYSPLIT uncertainties. The case shown in Sect 4, was already discussed (based on FLEXPART) in detail in the Hofer et al. (2017) paper.

Conclusions. (page 10, lines 14-17). This statement is confusing as written. The authors first they suggest to use globally valid conversion factors and then recommend to use regional ones. Maybe each suggestion should be followed with an uncertainty estimate. Please consider to rephrase the recommendations, since these are the ones to be followed by a potential user of POLIPHON.

Improved

---

## Author Comment (AC3) · 5 Jul 2019

The comment was uploaded in the form of a supplement:
https://www.atmos-meas-tech-discuss.net/amt-2019-98/amt-2019-98-AC3-supplement.pdf